# Compact A15 Frank-Kasper nano-phases at the origin of dislocation loops in face-centred cubic metals

Alexandra M. Goryaeva [1] ✉, Christophe Domain [2], Alain Chartier[3], Alexandre Dézaphie[1,4], Thomas D. Swinburne [5], Kan Ma[1,6], Marie Loyer-Prost[1], Jérôme Creuze[4] & Mihai-Cosmin Marinica [1] ✉

It is generally considered that the elementary building blocks of defects in face-centred cubic (fcc) metals, e.g., interstitial dumbbells, coalesce directly into ever larger 2D dislocation loops, implying a continuous coarsening process. Here, we reveal that, prior to the formation of dislocation loops, interstitial atoms in fcc metals cluster into compact 3D inclusions of A15 Frank-Kasper phase. After reaching the critical size, A15 nano-phase inclusions act as a source of prismatic or faulted dislocation loops, dependent on the energy landscape of the host material. Using cutting-edge atomistic simulations we demonstrate this scenario in Al, Cu, and Ni. Our results explain the enigmatic 3D cluster structures observed in experiments combining diffuse X-ray scattering and resistivity recovery. Formation of compact nano-phase inclusions in fcc structure, along with previous observations in bcc structure, suggests that the fundamental mechanisms of interstitial defect formation are more complex than historically assumed and require a general revision. Interstitial-mediated formation of compact 3D precipitates can be a generic phenomenon, which should be further explored in systems with different crystallographic lattices.

Structural defects impact the intrinsic properties of crystalline materials. The kinetics and interaction of defects control the microstructural evolution and, over time, it can significantly alter material properties[1–4]. For over sixty years, it has been generally accepted that the elementary building blocks of defects, like interstitial dumbbells and vacancies, cluster into 'bundles', which then aggregate by diffusion and form 2D dislocation loops[5]. Whilst numerous transmission electron microscopy (TEM) studies of irradiated metals evidence the presence of nanometric dislocation loops[1,6–8], there is still no direct experimental evidence that elementary building blocks arrange directly in 2D structures forming a nucleus of a dislocation loop. In this work, we revisit the conventional scenario of interstitial loop formation in fcc metals and investigate the competition between 2D interstitial clusters and compact 3D nano-phase inclusions with A15 Frank-Kasper phase structure[9]. Discovered almost 90 years ago[10], the compact A15 Frank-Kasper phase (space group Pm3n) is prevalent for $A_3B$ intermetallic alloys[11]. The only pure metal known to adopt the A15 structure is W[10]. This metastable allotrope is also called β-W.

In fcc metals, large vacancy- and interstitial-type loops have been extensively studied and characterised in TEM. For both types, the population is composed of faulted $\frac{1}{3}\langle 111 \rangle$ Frank loops[7] and prismatic $\frac{1}{2}\langle 110 \rangle$ dislocation loops[6]. The competition between Frank and

[1]Université Paris-Saclay, CEA, Service de recherche en Corrosion et Comportement des Matériaux, SRMP, Gif-sur-Yvette 91191, France. [2]EDF-R&D, Département Matériaux et Mécanique des Composants (MMC), Les Renardieres, Moret sur Loing Cedex F-77818, France. [3]Université Paris-Saclay, CEA, Service de recherche en Corrosion et Comportement des Matériaux, Gif-sur-Yvette 91191, France. [4]Université Paris-Saclay, ICMMO/SP2M, UMR 8182, Orsay 91405, France. [5]Aix-Marseille Université, CNRS, CINaM UMR 7325, Campus de Luminy, Marseille 13288, France. [6]School of Metallurgy and Materials, University of Birmingham, Birmingham B15 2TT, UK. ✉e-mail: alexandra.goryaeva@cea.fr; mihai-cosmin.marinica@cea.fr

prismatic loops is mainly governed by intrinsic properties, such as elasticity, stacking fault energies, and short-range chemistry, as well as by external conditions, like temperature and pressure[5,12]. While Frank loops are intrinsically immobile, the high mobility of prismatic loops promotes their elimination from surfaces, which reduces the density of these defects in TEM samples[6].

In contrast to the well-established behaviour of large defects observable in TEM, the morphology and nucleation process of small defects in fcc metals is still obscure. In the case of vacancies, the elementary defects diffuse and aggregate in voids and stacking fault tetrahedra (SFTs)[13–15]. Many experimental and theoretical studies[14,16–18], indicate that planar vacancy clusters rearrange into SFTs by assembling $\frac{1}{3}\langle 111\rangle$ Frank partial dislocations. Other works, based on atomistic simulations, suggest that voids can directly transform into SFTs[19,20].

The formation mechanism of small interstitial-type clusters and their growth is still not clear. The conventional 2D growth scenario[5,21,22] suggests that $\langle 100\rangle$ dumbbells arrange directly in 2D clusters, and form dislocation loops. However, theoretical studies of self-interstitial atoms (SIAs) agglomeration into clusters[12,23–28] do not provide clear confirmation or contradiction of direct 2D clustering. While most of the studies of SIA clusters in fcc Cu focus exclusively on 2D morphologies[12,27,28], an early study by Ingle et al.[23] suggests that SIAs in Cu can arrange into small 3D clusters with icosahedral structures. Moreover, research studies from the late 1970s to early 1980s, based on resistivity recovery experiments and diffuse scattering, indicate that the appearance of dislocation loops in fcc Al[29,30], Cu[31], and Ni[31–33] is preceded by small 3D clusters. However, the structure of these clusters has remained unknown, being considered a stochastic agglomeration of $\langle 100\rangle$ dumbbells and $\frac{1}{2}\langle 110\rangle$ crowdions. The existence of those clusters and their role in the evolution of microstructure were systematically neglected in later studies.

In this work, we investigate the mechanisms of SIA cluster nucleation and their subsequent evolution in fcc Al, Cu, and Ni. For all three metals, we use DFT calculations to compare the relative stability of A15 inclusions and conventional 2D cluster structures. By the means of ab initio calculations we have reinterpreted experimental studies of fcc metals, which combine resistivity recovery and diffuse X-ray scattering. For all three investigated metals, we obtain remarkable agreement with the experiments and demonstrate that 3D clusters in experimentally observed fcc are A15 nano-phase clusters. Moreover, we have designed an experimental setup based on electron-irradiated fcc Ni dilute alloy, which emphasizes the consequence of A15 clusters' existence on the morphology of large dislocation loops. Further, the formation of A15 inclusions is investigated using large-scale simulations of irradiation damage. The A15 clusters of critical size detected in these simulations are then used for the systematic exploration of the potential energy landscape and the investigation of transformation mechanisms between the A15 clusters and dislocation loops. The

presentation of simulation results is followed by a discussion of important implications and perspectives.

## Results

### Formation of A15 interstitial clusters and their relative stability with respect to dislocation loops

The accumulation of defects in irradiated materials is a stochastic process, driven by the structural rearrangement of atoms in a particular metastable basin of the energy landscape. In this section, by means of DFT calculations, we determine the relative stability of the A15 clusters with respect to the conventional 2D SIA loops and describe their formation mechanism.

Figure 1 depicts the formation of 3D A15 clusters in fcc metals from the elementary building blocks. Small A15 clusters ($N \leq 7$) are formed via the accumulation of non-parallel (mutually-orthogonal) $\langle 100\rangle$ dumbbells around the interstitial atom at the octahedral site. The dumbbells are located in the centres of the faces of fcc unit cell and form an icosahedron, whilst the interstitial atom at the octahedral site centre this icosahedron. Accumulation of seven interstitial atoms $I_7^{A15}$ (six dumbbells and one octahedral interstitial) yields a complete centred icosahedron. This configuration corresponds to the unit cell of $\beta$-W structure. Bigger A15 clusters ($N > 7$) can be formed by further agglomeration of interstitial atoms in fcc octahedral sites around the A15 icosahedron (e.g., $I_{13}^{A15}$ structure in Fig. 1), followed by subsequent accumulation of dumbbells that built icosahedral structures. Alternatively, some low energy A15 configurations with $N > 7$ (e.g., $I_{11}^{A15}$ cluster in Fig. 1) are formed by incomplete icosahedra sharing edges along $\langle 100\rangle$ direction. Overall, the formation process of A15 clusters in fcc matrix occurs through the progressive accumulation of interstitial atoms at fcc octahedral sites and $6c$ Wyckoff positions of $\beta$-W structure with an energetic preference for most compact cluster configurations.

The size-dependent formation energies and relative stability of different SIA clusters in fcc Al, Cu, and Ni are reported in Fig. 2. In this study, we disregard $\langle 100\rangle$ 2D loops as they have much higher formation energies[5]. In all three metals, the small-size A15 structures are in competition with low-energy 2D clusters (Fig. 2), i.e., with Frank dislocation loops $\frac{1}{3}\langle 111\rangle$ in Al and Ni, and prismatic $\frac{1}{2}\langle 110\rangle$ loops in Cu. With increasing defect size, the formation energy of A15 increases faster than that of dislocation loops, since it scales as $N$ (like an Eshelby's inclusion) for A15 and as $\sqrt{N}\ln(N)$ for 2D loops[18,34]. This yields large-size A15 clusters less stable than 2D loops. Based on our ab initio simulations, among the three investigated metals, the biggest A15 clusters can be expected to form in Ni (Fig. 2f), where they can potentially attain a critical size of 10-12 SIAs. The 3D clusters in Al and Cu are likely to be smaller and their maximal size may reach 7-8 SIAs in Al (Fig. 2d) and 8-10 SIAs in Cu (Fig. 2e).

Some early studies of small SIA clusters in fcc Cu[23,28], based on Embedded Atom Method (EAM) potentials, have considered a possible

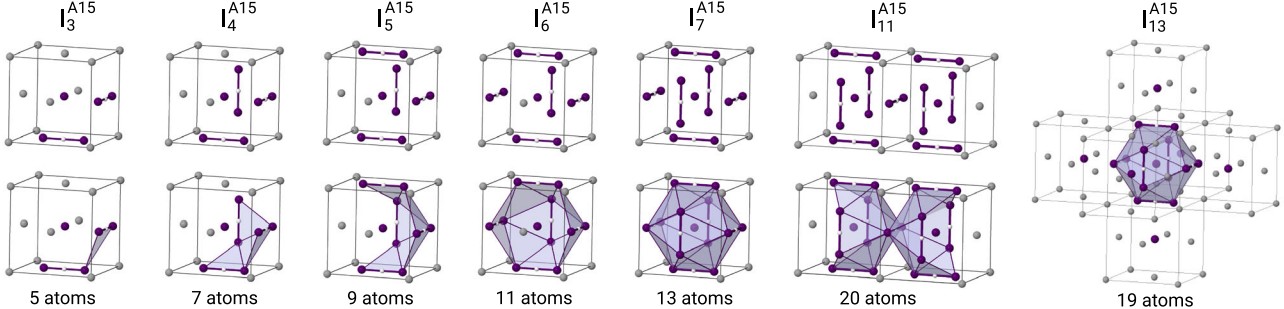

**Fig. 1 | Formation of A15 nano-phase inclusions in fcc metals.** The 3D clusters $I_N^{A15}$ with $N$ SIA nucleate through accumulation of mutually-orthogonal $\langle 100\rangle$ SIA dumbbells and interstitial atoms at the octahedral site. The atoms of fcc and A15 structures are shown in grey and purple, respectively. For each configuration, the total number of atoms that contribute to the A15 cluster is indicated at the bottom of each structure. The depicted structures correspond to the low energy configurations from Fig. 2.

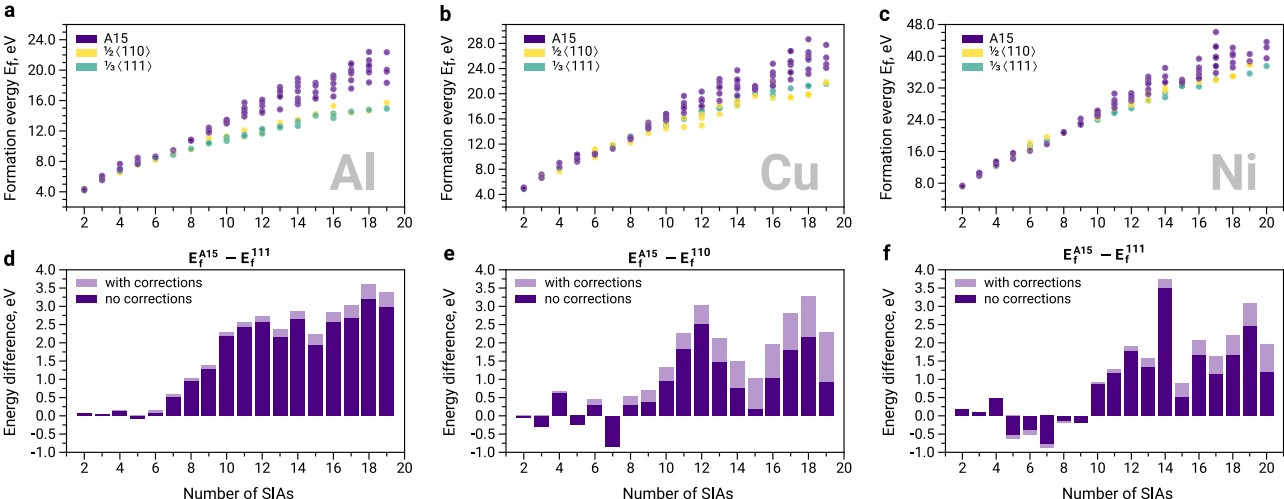

**Fig. 2 | Formation energies and relative stability of A15 clusters and interstitial dislocation loops in fcc metals.** Subplots (**a**–**c**) provide the formation energies in Al, Cu, and Ni obtained in DFT calculations. Subplots (**d, g, h**) report the relative stability of A15 clusters with respect to the most stable dislocation loop family in each metal: $\frac{1}{3}\langle 111 \rangle$ in Al (**d**) and Ni (**f**), and $\frac{1}{2}\langle 110 \rangle$ in Cu (**e**). The negative energy difference in (**d**–**f**) indicates that A15 clusters are more stable than the 2D clusters. Light purple colour in (**d**–**f**) denotes the energy differences that take into account elastic corrections (see Supplementary Note 2 for more details); non-corrected energy differences are shown in dark purple.

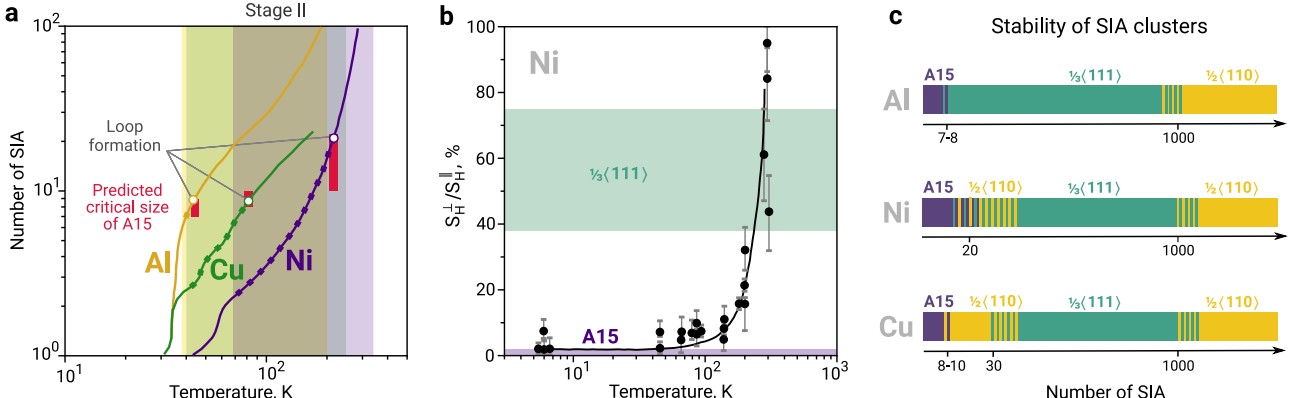

**Fig. 3 | Growth of interstitial clusters in fcc metals. a** The experimental findings in diffuse X-ray scattering and resistivity recovery experiments in Al[29,30], Cu[31] and Ni[31–33] compared with the theoretical predictions of this work. The average size of clusters (in the number of SIAs) determined in experiments is reported as a function of temperature, while our predictions are based on 0 K DFT simulations. The curves for Al, Cu, and Ni are shown in yellow, green, and purple, respectively. For each metal, the interval of Stage II is indicated with a similar colour. The experimental data are taken from ref. 35. The open circles indicate the experimental size-temperature estimation of the dislocation loop formation. Red rectangles report the critical size of A15 clusters from our DFT simulations. **b** The experimental measurement in Ni, from ref. 32, of the ratio of $S_H^{q,\perp}(\mathbf{k})/S_H^{q,\parallel}(\mathbf{k})$ (see Methods) of the Huang scattering in the directions perpendicular and parallel to **q**. The coloured rectangles in purple - for A15 nano-phase clusters - and green - for $\frac{1}{3}\langle 111 \rangle$ Frank loops - are the present theoretical estimation of the range of the previously defined scattering ratio from the DFT values of elastic dipole tensor $P_{ij}$ of various defect morphologies having sizes between 2 and 25 SIAs clusters. **c** Schematic illustration of interstitial-type defects stability in fcc Al, Cu, and Ni. The cluster size is provided in number of SIAs. The dashed areas indicate the size of clusters for which metastability phenomena can manifest, giving rise to the coexistence of at least two defect types.

3D arrangement of non-parallel $\langle 100 \rangle$ dumbbells. Although those studies found that 3D clusters are more stable than 2D defects for $N < 9$, their presence in fcc materials was systematically disregarded in later works. The majority of SIA studies[12,21,27,28] were focused on the relative stability of 2D loop morphologies reported by experimental observations, i.e., $\frac{1}{3}\langle 111 \rangle$ and $\frac{1}{2}\langle 110 \rangle$. Here we find that the basin of small 3D clusters with A15 structure is not far in energy from the basin of 2D dislocation loops. In other words, small A15 clusters in Al, Cu and Ni can form together with 2D SIA structures and play an important role in the formation and growth of interstitial-type defects in fcc materials.

### Evidence of the A15 clusters formation from experiments

Clusters with sizes of 7-10 SIAs, as we observe in the DFT simulations (Figs. 1, 2), can be investigated in sophisticated experiments that combine in-situ observations of diffuse X-ray scattering or diffuse Huang scattering and the resistivity recovery experiments. We give special attention to the first two stages and particularly consider the end of Stage II, where large interstitial clusters are formed. In Fig. 3a, we summarize the experimental findings on the size and type of defects formed in resistivity recovery experiments in Al, Cu, and Ni, as reported by Ehrhart et al.[35], and compare them with our theoretical predictions. For all considered metals, there is a common trend in the formation of small SIA clusters. At the end of Stage I, small clusters of two to four SIAs are formed. Then, during Stage II, larger clusters are observed. Diffuse X-ray scattering indicates that these clusters have a 3D structure with up to six to seven SIAs in Al and Cu, and twenty to thirty in Ni. For bigger cluster sizes, the same experimental method indicates the appearance of dislocation loops.

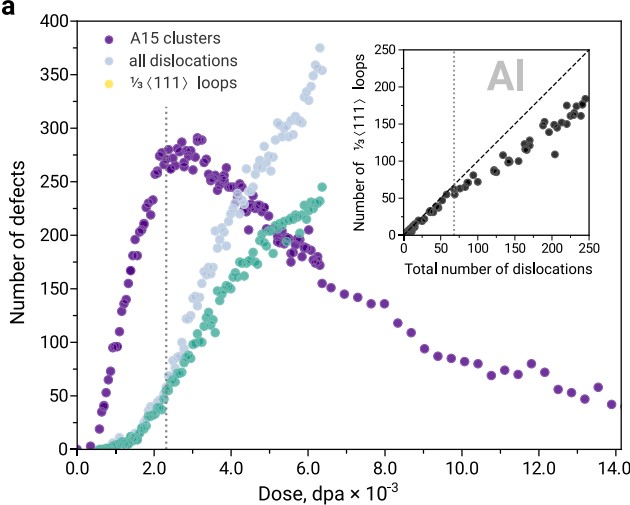

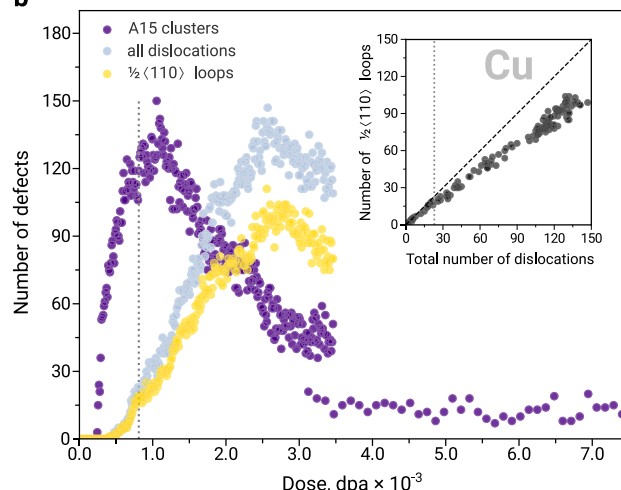

**Fig. 4 | Evolution of A15 clusters and dislocation loops with dose in Frenkel pair accumulation simulations.** Number of A15 clusters and dislocations in (**a**) fcc Al and (**b**) fcc Cu as a function of dose in a simulation cell with 864,000 atoms at 300 K.

The experimentally observed crossover size in Al and Cu are in very good agreement with our theoretical predictions based on the 0 K DFT energy landscape (Fig. 3a). For Ni, the theoretical simulations suggest a crossover of 10-12 SIAs, while the experiment results provide much larger crossover values between twenty and thirty SIAs. This discrepancy may originate from thermal or magnetic effects, which are not possible to identify in the framework of the present study.

Further, we consider in detail the case of Ni of the experimental study of Bender et al.[32] that provides experimental values of average defect size by recording simultaneously the resistivity recovery and diffuse Huang scattering experiments. Having the accurate values of the defect dipole tensor $P_{ij}$ from DFT calculations of A15 and dislocation loops up to 25 SIAs, we are able to reinterpret those experiments. The measurements in ref. [32] were performed over the first three stages of fcc Ni irradiated with electrons: Stage I before 70 K, Stage II between 70 K and 300 K, and Stage III between 300 K and 500 K. Within the first stage, $I_{D,E}$ SIAs begin to agglomerate, and at the end of Stage I, small clusters of $\langle n \rangle$ = 2-3 interstitials are formed. During Stage II, these clusters slowly grow at 70 K < T < 200 K, and an important increase in cluster size is observed between 200 K and 300 K. The size of the clusters is estimated by the intensity of Huang scattering and the resistivity values at a given temperature. The geometry of defects (3D versus 2D) is deduced by the characteristic values of scattering function $rS_H = S_H^{\perp}/S_H^{\parallel}$ (as in Fig. 3b reproduced from[32]). This function $rS_H$ can be computed for each defect from the elastic constants of the material and the accurate values of the elastic dipole tensor of the defect (see Methods). Figure 3b reports the values of $rS_H$ function for A15 and $\frac{1}{3}\langle 111 \rangle$ loops. For each type of defect, we compute $rS_H$ function at the conditions given by the experiment[32] (i.e., **q** = [010] direction at the point of reciprocal lattice **G** = (400) using the convention described in Methods).

The theoretically predicted values (Fig. 3b) of $rS_H$ for A15 clusters are close to zero, while for $\frac{1}{3}\langle 111 \rangle$ they range between 40% and 60% (in general proportional to the size). The values associated to perfect loop $\frac{1}{2}\langle 110 \rangle$ are much larger, around 200%. The most important aspect is that there is a gap in values of $rS_H$ between A15 and $\frac{1}{3}\langle 111 \rangle$ loops. There are no intermediate values between 0% and 40%, regardless of the SIAs cluster size. The Huang scattering signal is proportional to the density of various morphologies of SIAs (e.g., see the Eq. (2) in the Methods section). The intermediate experimental values around 20−30% of $rS_H$ come from the average between the almost zero signal of A15 3D clusters and the signal of dislocation loops, which start to form from A15 clusters at the end of Stage II. Thus, Fig. 3b suggests that for small

SIA clusters, there are no defects other than A15 clusters, as long as the $rS_H$ signal is close to zero. The above results unambiguously demonstrate that the 3D defects predicted by the experiments are clusters that are energetically stable and kinetically trapped in the attraction basin of the Frank-Kasper A15 nano-phase with an almost zero $rS_H$ signal.

In the following section, we will use large-scale atomistic simulations to investigate the nucleation and transformation of A15 clusters in irradiated Al and Cu, the metals where theory and experiment consistently suggest the transformation around 7-10 SIAs.

## Evidence of the A15 clusters formation from the large-scale calculations of radiation damage

In order to investigate the role of A15 clusters in the formation of interstitial-type defects in fcc metals, we perform large-scale calculations of irradiation damage. Simulations of Frenkel pair accumulation (FPA) is particularly suitable for extensive exploration of defect populations, as well as of the related processes dominated by short-range diffusion[3,4,36]. Here we employ FPA as a tool to explore the onset of SIA cluster formation and the appearance of first dislocation loops. To complement the FPA simulations, we also perform calculations of displacement cascades, as they take into account long-range, i.e., thermal, diffusion processes (see Methods). The simulations in fcc Al and Cu are performed using EAM potentials that are numerically fast and allow for good qualitative agreement with DFT calculations (see Supplementary Note 1). For fcc Ni, there is no suitable semi-empirical potential, therefore, no large-scale simulations were performed for this material.

The FPA calculations evidence the massive formation of small A15 clusters in Al and Cu at the very beginning of the irradiation process. In both fcc metals, we did not observe any A15 clusters bigger than 20 atoms, which corresponds to the maximum number of 14 interstitial atoms. Figure 4 provides the number of formed interstitial-type defects as a function of dose and illustrates the evolution of the microstructure. The simulations suggest that nucleation of A15 clusters in both materials occurs prior to the formation of dislocation loops.

The phenomenon of A15 clusters formation prior to dislocation loops is particularly interesting in the case of Al where small A15 clusters ($N \leq 7$) are expected to be metastable with respect to small Frank loops (Fig. 2c). The observed A15 formation is favoured by the specific morphology of icosahedral clusters, which can be attained by agglomeration of $\langle 100 \rangle$ dumbbells via the low-energy translation-

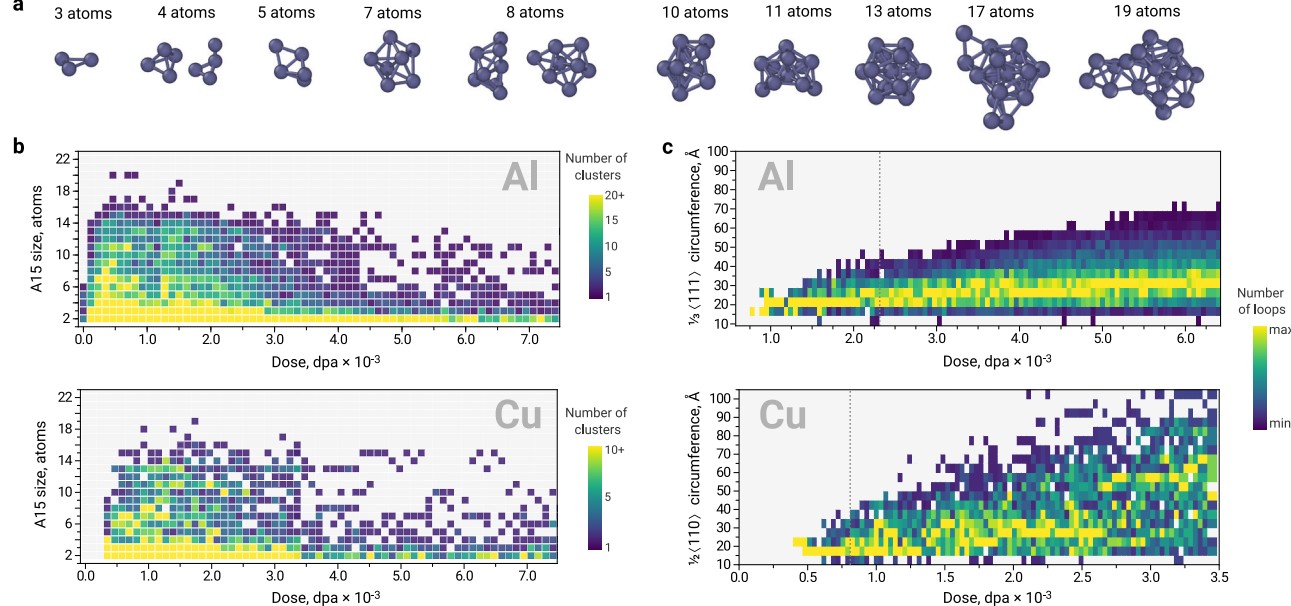

**Fig. 5 | Size distribution of A15 clusters and dislocation loops from Frenkel pair accumulation simulations. a** The typically observed A15 clusters with sizes from 3 to 19 atoms. The size refers to the total number of atoms that contribute to the A15 cluster, as indicated at the bottom of Fig. 1. **b** Evolution of A15 cluster size as a function of dose in fcc Al and Cu. **c** The size of Frank $\frac{1}{3}\langle 111\rangle$ loops in fcc Al and prismatic $\frac{1}{2}\langle 110\rangle$ loops in fcc Cu. The reported sizes correspond to the loop circumference defined by DXA analysis[70]. For each dose, the colour code is applied such as the minimum and the maximum number of the loops always correspond to purple and yellow, respectively. The grey dotted lines indicate the limit of the regime where the loops are the only dislocation defects formed in Al and Cu, as defined in Fig. 4.

rotation mechanism (ca. 0.1 eV). The formation of $\frac{1}{3}\langle 111\rangle$ loops requires the translation of dumbbells in combination with their rotation from $\langle 100\rangle$ to $\langle 111\rangle$ directions, which requires over 0.5 eV energy. Thus, the simple mechanism of A15 formation from low energy $\langle 100\rangle$ dumbbells favours the formation of metastable 3D clusters prior to 2D SIA platelets. The distribution of A15 cluster sizes in Al (Fig. 5b) resembles the shape of a Poisson distribution with a maximum at low doses. This distribution is not biased by attractive configuration and samples a pure metastable basin (Fig. 2c). At the early stages of irradiation (the regime with the limit of $2.3 \times 10^{-3}$ dpa, indicated with grey dashed line in Fig. 4a and its inset plot), only Frank dislocation loops are formed in Al. Interestingly, the majority of Frank loops formed in this regime have a circumference of ca. 20–21 Å (Fig. 5c). Thus, instead of gradually increasing in size, many interstitial Frank loops appear at the size of 7–8 interstitial atoms. This size is consistent with the critical size of A15 clusters (Fig. 2b, c), suggesting that A15 clusters transform into $\frac{1}{3}\langle 111\rangle$ dislocation loops (Fig. 4a) after reaching the critical size. In order to better tackle a possibility of Frank loop nucleation from A15 clusters in Al, we investigate the transition channels of $I_7^{A15} \to I_7^{\langle 111\rangle}$ using Activation-Relaxation Technique nouveau (ARTn)[37] and Nudged Elastic Band (NEB) calculations[38]. Figure 6a depicts the identified transition mechanism with the lowest energy saddle point between the attraction basin of A15 clusters and $\frac{1}{3}\langle 111\rangle$ loops. This transition barrier of the perfect icosahedral $I_7^{A15}$ cluster into the Frank loop is nearly 0.5 eV (green curve in Fig. 6a). The transformation, presented in Fig. 6d, occurs via $\langle 111\rangle$ screw mechanism (rotation around the $\langle 111\rangle$ axis combined with the displacement along the $\langle 111\rangle$ direction).

With increasing dose, the nucleated Frank dislocation loops in Al grow bigger and interactions between the loops initiate the formation of Shockley partials $\frac{1}{6}\langle 112\rangle$ and other dislocation types. From a dose limit of $4.2 \times 10^{-3}$ dpa, the number of A15 clusters drastically decreases (Fig. 4a), and the evolution of the microstructure is mainly controlled by the dislocations. The developed dislocation network absorbs newly generated interstitial atoms, which significantly diminishes the nucleation rate of A15 clusters compared to the early stages of the irradiation process.

In fcc Cu, the number of interstitial A15 clusters is almost twice smaller than in Al. This observation is consistent with the previous studies that report a low tendency of interstitial-type defects to cluster in this material[39,40]. The distribution of A15 cluster size in Cu is noisier than in Al. However, one can distinguish a preference for the formation of defects with particular sizes, e.g., 13 atoms, which is consistent with the stability of $I_7^{A15}$ clusters (Fig. 2, Supplementary Note 1). Similarly to Al, the majority of formed dislocation defects at the beginning of irradiation (the limit of this regime is $0.81 \times 10^{-3}$ dpa, indicated with grey dashed line in the Fig. 4b and its inset plot) are loops. However, in contrast to Al, these are $\frac{1}{2}\langle 110\rangle$ prismatic loops. As in Al, the limit of the regime where only loops of a certain type are formed coincides with the maximum of A15 concentration in the material. The majority of prismatic loops formed in this regime in Cu have a circumference of 17–18 Å and 21 Å (Fig. 5c), which corresponds to 8–13 SIAs. The exclusive formation of certain loop types with consistent size in Cu till $0.81 \times 10^{-3}$ dpa suggests that the formation of these loops does not occur by a gradual accumulation of SIA dumbbells into 2D structures but by an instant formation process. The observed size of dislocation loops is consistent with the predicted critical size of A15 clusters in fcc Cu (Fig. 2, Supplementary Fig. 1). The transition barriers of $I_8^{A15}$ clusters in Cu and their energy landscape identified using TAMMBER[41] and NEB[38] calculations are provided in Fig. 6b, c. The transition barriers of $I_8^{A15}$ clusters into $\frac{1}{2}\langle 110\rangle$ loops are in the range of 0.5–0.8 eV, close to those for transition into $\frac{1}{3}\langle 111\rangle$ loops in Al (Fig. 6a).

In order to complement the FPA calculations, we further consider the displacement cascades (see Supplementary Note 3) where long-range diffusion takes place. The number of A15 clusters detected both in Al and Cu per cascade is small and represents only 4–5% of interstitial atoms, which is consistent with the number of 3D clusters with C15 structure in bcc Fe produced in displacement cascades[42,43]. The C15 clusters in bcc Fe, which form only 5% of SIAs clusters in a single cascade, were shown to have a major impact in the overlapping cascades[43]. In perspective, it will be interesting to explore the effect of A15 clusters in fcc metals on the population of dislocation loops in overlapping cascades.

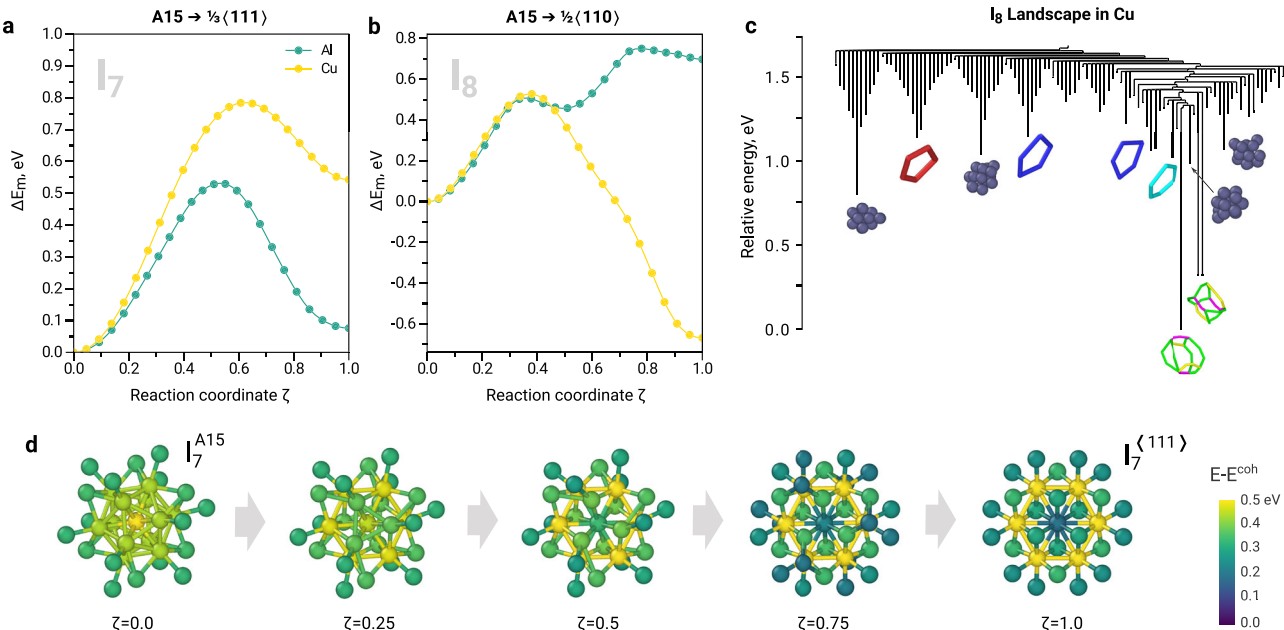

**Fig. 6 | Transition of A15 interstitial clusters into 2D loops. a** Transition barriers of $I_7^{A15}$ clusters to $\frac{1}{3}\langle 111\rangle$ Frank loops in Al and Cu. **b** Transition barriers of $I_7^{A15}$ clusters to $\frac{1}{2}\langle 110\rangle$ prismatic loops in Al and Cu. The initial and final states in (**a**, **b**) were found using the Activation-Relaxation Technique nouveau (ARTn)[37]. **c** Truncated disconnectivity graph between different states of $I_8$ clusters in fcc Cu, as discovered by TAMMBER[41,66]. A15 clusters are detected using distortion scores[63]; dislocation lines are identified using DXA analysis[70]. A15 clusters are shown with purple atoms, $\frac{1}{3}\langle 111\rangle$ faulted Frank loop is cyan, $\frac{1}{2}\langle 110\rangle$ loops are blue, $\frac{1}{6}\langle 411\rangle$ loop is red, $\frac{1}{6}\langle 112\rangle$ Shockley dislocations are green. The lowest energy dislocation configurations are built by partial Shockley dislocations that form perfect prismatic loops according to the reaction $\frac{1}{6}\langle 211\rangle + \frac{1}{6}\langle 12\bar{1}\rangle \rightarrow \frac{1}{2}\langle 110\rangle$. **d** The transition mechanism of the perfect icosahedral A15 cluster to the Frank loop. The atoms are coloured according to the local atomic energies. The corresponding energy barrier is shown with the green curve in subplot (**a**). The structures are viewed along the $\langle 111\rangle$ direction.

## Discussion

### Morphology of interstitial clusters: an interplay between equilibrium and metastability

Our atomic-scale simulations emphasise that SIA dumbbells in fcc metals tend to first cluster into metastable 3D compact inclusions with A15 structure instead of directly forming 2D clusters. When reaching the critical size, these 3D clusters eventually transform into either Frank or prismatic loops (Fig. 3c), depending on the relative stability of defects in the material, and act as a primary source of dislocation loops in irradiated Al, Cu, and Ni. Here, we discuss the present findings in the context of the relative stability of the defect clusters governed by the interplay between the elasticity and metastability of small clusters.

At small sizes, interstitial clusters do not follow the standard elastic theory[18], which applies to large dislocation loops. The main contribution to the energy of defect is given by chemical bonding and local atomic arrangement of interstitial atoms, whilst the elastic energy is given by the elastic interaction between the defect and the matrix (e.g., elastic dipole energy[44]). The theoretically predicted crossover size between A15 and the dislocation loops is in very good agreement with experimental observations[21]. This indicates that relative stability from 0 K DFT simulations is a good indicator for the critical size of A15 in Al and Cu. However, in the case of Ni, the agreement is not so remarkable, which potentially can be explained by finite temperature effects, such as vibrational, magnetic, or kinetic, which should be investigated in future studies.

Under extreme conditions, the three basins of small interstitial-type defects in fcc metals are filled stochastically (e.g., by the insertion of Frenkel pairs - molecular dynamics - atomic relaxation cycles of FPA) and do not necessarily follow the equilibrium statistics by occupying the most energetically favourable basin firstly. In this work, we demonstrate that metastable states, such as A15, can be filled first. The formation of A15 clusters is favoured by their intrinsic geometry, which can be easily created via low energy (ca. 0.1 eV) diffusion mechanism of $\langle 100\rangle$ dumbbells[22,45]. Although being a metastable defect family, (Fig. 2), the basin of small A15 clusters is filled prior to dislocation loops (Figs. 4a, b, 3) both in Al and Cu.

Another interesting case in this work is given by the relative stability of small loops in Cu, where prismatic $\frac{1}{2}\langle 110\rangle$ loops are the most stable configurations (Figs. 2e, 3), infringing the common belief based on the elastic theory (see Supplementary Note 5). In contrast to Al, A15 clusters in Cu transform into $\frac{1}{2}\langle 110\rangle$ prismatic loops, which are expected to be stable up to ca. 30 SIA (see Supplementary Note 1). However, the direct transformation of $\frac{1}{2}\langle 110\rangle$ into $\frac{1}{3}\langle 111\rangle$, e.g., by on-site rotation of interstitial atoms or by fault creation through reactions of dislocations network, is energetically costly. Consequently, even for large defect sizes, dislocations can maintain glissile $\frac{1}{2}\langle 110\rangle$ morphology and be eliminated from the system on attractive sinks before the transformation into low energy Frank loop. Being impossible to eliminate $\frac{1}{2}\langle 110\rangle$ loops in our FPA simulations, the system develops dislocation reactions in order to form Frank loops from prismatic loops, in agreement with the energetic landscape of defects.

Due to the small size of A15 clusters, their direct experimental evidence is challenging. In order to illustrate how A15 clusters can increase the degree of stochasticity and favour the formation of various defect morphologies, we have designed an in-situ transmission electron microscopy (TEM) experiment, which monitors the appearance and growth of dislocation loops in electron-irradiated Ni-0.4Ti. Figure 7 reports the in-situ evolution of interstitial-type defects from the stage of black dots till the appearance of detectable dislocation loops. Without metastable A15 clusters, one can expect the formation and gradual growth of only one family of dislocation loops. However, the four black dots of interstitial character have transformed into three perfect loops $\frac{1}{2}\langle 110\rangle$ and one Frank loop, which may result from the stochastic character of filling the energetic basins.

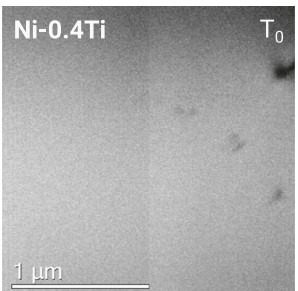
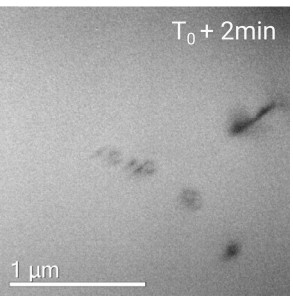
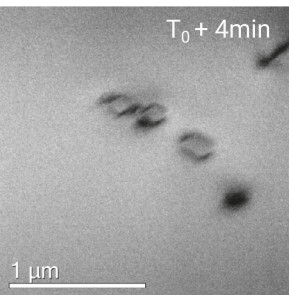
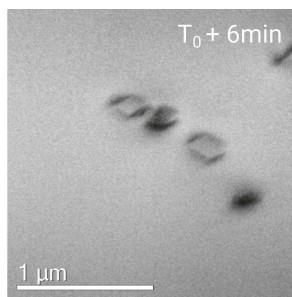

**Fig. 7 | Growth of interstitial dislocation loops in fcc Ni-0.4Ti.** TEM micrographs of in-situ electron irradiation recorded under two-beam Kinetics Bright Field conditions using **g** = ⟨200⟩.

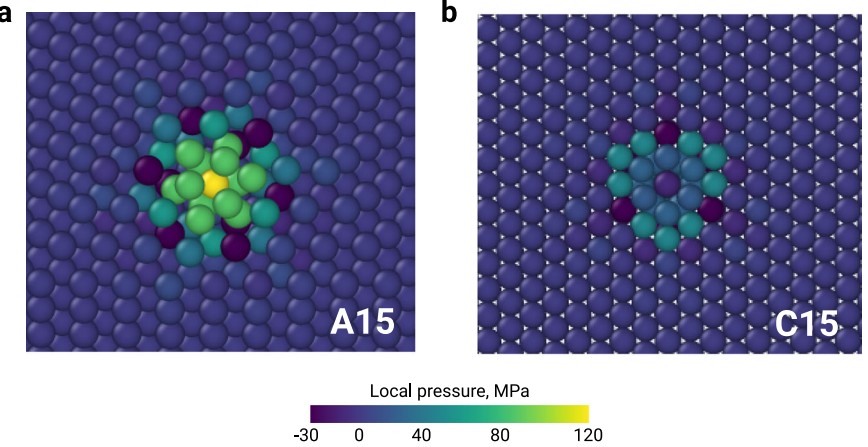

**Fig. 8 | Comparison of the A15 cluster in fcc Al with the C15 cluster in bcc Fe.**
**a** Local pressure in the A15 cluster that builds a complete icosahedron with 7 interstitial atoms in fcc Al. **b** Local pressure in the C15 cluster that builds a complete Z16 polyhedron in bcc Fe with 12 interstitial atoms and 10 vacancies. The clusters are computed using the EAM potentials by Mendelev et al.[46] for Al and by Marinica et al.[42] for Fe. The structures are shown in the {111} plane and cut such as the atoms centring the polyhedra are visible.

## 3D interstitial clusters in fcc and bcc lattices: A15 versus C15

Creation of interstitial dislocation loops modulated by an intermediate step of 3D cluster formation was previously observed in bcc Fe[4,34,42,43] where nano-phase precipitates with C15 Laves phase structure form prior to dislocation loops[4]. However, the structure and the behaviour of C15 clusters is different from those of the A15 clusters. A15 clusters are formed by icosahedral units, while C15 clusters are built by Z16 Frank-Kasper polyhedra. Structurally, the C15 clusters are formed by putting together vacancies and SIAs, e.g., a di-interstitial C15 cluster, which yields a complete Z16 Frank-Kasper polyhedron, is formed by 10 vacancies and 12 interstitial atoms. Thus, with $N$ SIA atoms, a precipitate of the C15 Laves phase can be formed by replacing $2N$ bcc atoms with $3N$ C15 atoms, whilst creating the A15 clusters requires inserting $N$ atoms without putting any vacancies. A complete A15 icosahedron is built by 7 interstitial atoms. As a consequence, the A15 defects are much more compact and accommodate up to 3 times higher local pressure in the centre of the cluster (Fig. 8) than the C15 clusters.

The compact nature of the A15 clusters with the local concentration of pressure within the defect plays an important role in the development and transformation of the A15 clusters under irradiation. Compared to C15, A15 clusters form and disappear faster. The vast majority of A15 clusters form at the early stages of irradiation and almost no 3D structures are present in the latter stages when the microstructure is dominated by the dislocation network (Fig. 4, Supplementary Fig. 5). Due to their intrinsic geometry, C15 clusters can form mixed structures with dislocation lines without being fully absorbed by dislocations[4,43]. Consequently, bcc Fe can reach a kinetic equilibrium between the C15 clusters and dislocations[4]. The flux of self-

interstitial atoms created by irradiation continuously contributes to the growth of both morphologies. A15 clusters in fcc metals behave differently. Due to the important concentration of pressure within the cluster, A15 fully transforms into loops. No defect configurations formed by A15 nano-phase clusters that are connected to dislocations or to dislocation networks were observed in our simulations.

## On the role of A15 clusters in microstructural evolution

Similarly to C15 clusters in bcc structure, A15 clusters in fcc structure are characterised by high cohesive energies. The migration barriers of ⟨100⟩ dumbbells in fcc Al and Cu (computed with EAM potentials[46,47]) are 0.11 eV and 0.09 eV, respectively. From ARTn simulations, we find that the barrier associated with the absorption of the dumbbell by a small A15 cluster is nearly one order of magnitude lower than the migration of ⟨100⟩ dumbbells. Such a low activation barrier indicates that the formation of $I_{n+1}^{A15}$ is very favourable from the reaction between $I_n^{A15}$ clusters and mobile dumbbells. This means that ⟨100⟩ SIAs will preferably contribute to the development of metastable A15 clusters and dislocation loops will appear only above a given size. This scenario of dislocation loop formation can alter our vision and understanding of the microstructural evolution, which historically takes into account only the continuous mechanism of loop formation, without any intermediate stages. Besides the size of defects, it is also important to consider the density of immobile 3D clusters in the matrix. In the case of bcc lattice where C15 clusters occur[4,34,42], recent multiscale studies of Fe and W[48–52] indicate that the presence of immobile clusters in matter drastically change the microstructural evolution, enhancing the accumulation of defects even at low doses.

In this context, the accurate characterisation of various defect morphologies, including 3D A15 nano-phase or 2D traditional loops, is of primary importance for the materials science community. If A15 clusters are not taken into account, characterized, and counted, the global formation volume balance between SIAs and vacancies, is incorrect and the large-scale predictions are inaccurate for such processes as swelling, which is critical in austenitic steels.

Finally, the present mechanism of dislocation loops formation via A15 clusters can bias the loop morphologies in the microstructure of materials under stress or in any strain field, such as thin foils or in the presence of a surface. The dipole tensor of A15 clusters has cubic symmetry (see Supplementary Note 6), i.e., it has no directional preferences, unlike that of dislocation loops[44,53–55]. This means that during the transformation from an A15 cluster to a dislocation loop, elastic interactions with the strain field will induce certain preferential habit planes of dislocation loops, which will correspond to the lowest elastic interaction energy[55,56]. Consequently, the mechanism of transformation A15 clusters → dislocation loops can impact the morphology of dislocation loops under strain, even at large sizes, which can be considerably biased.

## Perspectives

This work revises the historically assumed mechanism of interstitial defect formation in fcc metals. We argue that, instead of the direct formation of 2D clusters, interstitials first accumulate into 3D compact clusters with a structure similar to inclusions of A15 Frank-Kasper nano-phases in a host material. Our study suggests that A15 clusters are expected to occur in fcc Al, Cu, and Ni under irradiation with electrons (e.g., in TEM) and ions. These A15 clusters form prior to the planar defects and further act as a primary source of interstitial dislocation loops. The formation of 3D clusters is mainly governed by the geometry and general tendency of SIAs to cluster into dense structures that subsequently transform into dislocation loops. Accumulation of SIAs into 3D clusters was previously considered as a specific phenomenon of bcc Fe, where C15 clusters occur. Thus, 3D nano-phase inclusions likely represent a general phenomenon, which should be further explored in other metals, e.g., in hcp.

The theoretical prediction of compact A15 clusters elucidates the 3D cluster structures observed in experiments that combine diffuse X-ray scattering and resistivity recovery. In the case of Al and Cu, there is a remarkable agreement between theoretical and experimental predictions of the critical A15 size, near 7-10 SIAs. In the case of Ni, the 3D clusters are expected to reach bigger sizes up to 20-30 SIAs. The energetic basin of small A15 clusters in fcc Al and Cu is close to $\frac{1}{3} \langle 111 \rangle$ Frank loops and to $\frac{1}{2} \langle 110 \rangle$ prismatic loops, respectively. In Ni, the small A15 clusters can transform into both types of loops. The formation of metastable A15 clusters in irradiated Al, Cu, and Ni prior to any 2D clusters is related to the low energy migration barrier (ca $0.10 - 0.15$ eV[22,35]) of the most stable $\langle 100 \rangle$ dumbbells in these metals, which enables their easy agglomeration in the metastable basin on A15 clusters.

The present mechanism of transformation A15 clusters → dislocation loops is likely to impact the morphology of dislocation loops under stress or in any strain field, such as thin foils or in the presence of surfaces. In perspective, investigations of kinetic interactions of these 3D clusters with the interstitials and vacancies will allow quantifying their impact on the microstructural evolution in materials under extreme conditions.

The formation of A15 nano-phase clusters may have important implications for metallic alloys with a fcc lattice, such as austenitic steels and high-entropy alloys (HEAs). The stability of A15 clusters can be enhanced in fcc metals by including some dopants. In particular, adding elements that are known to form A15 intermetallic phases can be expected to favour the formation of A15 precipitates in fcc metals. Most commonly, the intermetallic A15 phases $A_3B$ are formed with $A$

atoms being transition metals: Ti, Zr, V, Nb, Ta, Cr, and Mo; $B$ atoms being the elements from the IIIB and IVB groups or the precious metals: Os, Ir, Pt, Au. Thus, for the Fe-Cr-Ni based alloys, like austenitic stainless steels or HEA studied in refs. [57,58], the presence of Si might favour the formation of $Cr_3Si$ precipitates with A15 structure. The known A15 phases $A_3B$ with $B$=Al have $A$=Nb, V, Mo[59,60]. For $B$=Cu, there are no known compounds of the A15 phase stable at ambient pressure[59]. In Au-based fcc alloys, A15 precipitates can be expected to form after alloying with Nb, Ti, V, or Zr[60]. Doping with Cr can potentially yield a formation of A15 clusters in fcc Ir, Pt, Rh[60].

## Methods
### Calculations of formation energies
The relative stability of A15 and 2D interstitial clusters is investigated in this work using DFT and EAM potentials calculations.

The DFT formation energies of the defect clusters (up to $N$ = 22) are computed using projector augmented wave (PAW) framework[61]. The employed electronic structures are $3s^2\,3p^1$ for Al, $3d^{10}\,4p^1$ for Cu, and $3d^8\,4s^2$ for Ni. The plane wave energy cutoff is 350 eV and the Hermite-Gaussian broadening width for Brillouin zone integration is 0.2 eV. The exchange-correlation energy is evaluated using the Perdew-Burke-Ernzerhof (PBE) Generalized Gradient Approximation (GGA). The simulation cells for $N$ SIAs contain between 500+$N$ ($5a_0 \times 5a_0 \times 5a_0$) and 1372+$N$ atoms ($7a_0 \times 7a_0 \times 7a_0$). The $k$-point grid mesh was chosen from $3^3$ for small cells up to (1 or $2^3$) for the largest cells. Each configuration is relaxed using the conjugate gradient with a convergence criterion on the force on each atom of 0.02 eV/Å. The calculations are performed at constant volume with the lattice parameter $a_0$ = 4.0397 Å for Al, $a_0$ = 3.522 Å for Ni, and $a_0$ = 3.635 Å for Cu. The computed formation energies were adjusted using the dipole correction[44]. The EAM formation energies of the defect clusters are computed using the potential by Mendelev et al.[46] for Al and by Mishin et al.[47] for Cu. For the clusters up to $N$ = 22, the simulation cells contain 13,500+$N$ atoms, while for the large clusters up to $N$ = 1000 are computed in the simulation cells with 1,372,000+$N$ atoms ($70a_0 \times 70a_0 \times 70a_0$).

### Large scale calculations of radiation damage
The radiation damage in fcc Al and Cu is studied using the Frenkel pair accumulation (FPA) and cascade calculations. The calculations are performed using the EAM potential by Mendelev et al.[46] for Al and by Mishin et al.[47] for Cu. These potentials provide a reasonable agreement with the DFT calculations of relative formation energies of small 2D interstitial loops with respect to the A15 clusters.

The FPA is a powerful tool for exploring complex energetic landscapes under electron irradiation at low temperatures. In order to mimic electron irradiation, the periodic creation of Frenkel pairs was first introduced by Limoge et al.[36] and has been further adapted for studies of irradiation-induced defects in different materials either at finite temperature or 0 K[3,4]. Due to the high irradiation flux (more than $10^6$ dpa/s) typical for FPA, establishing the direct link with the kinetic effects based on these simulations is impractical. Nonetheless, the method is particularly useful for exploring the morphology of defects and their density under irradiation. For example, the results of a high-flux FPA simulation can be applied to lower-flux overlapping cascades[3]. Furthermore, the FPA method allows for extensive exploration of the defect population and of the related processes dominated by short-range diffusion[3,4]. Moreover, FPA is particularly suitable for studies of interstitial-type defects, while other types of defects, such as vacancy migration with larger barriers, are blocked. In the present study, FPA calculations are performed in LAMMPS[62] in the fcc simulation cells with 864,000 atoms, where randomly chosen atoms are randomly displaced such as their distance from any other atom is not smaller than 1.4 Å. We tested also different cutoffs, namely 1.0, 1.2, and 1.4 Å and the results obtained do not differ from each other. 200 Frenkel

pairs are introduced every 2 ps. In order to control the pressure and temperature changes caused by the increasing number of interstitial atoms and vacancies in the simulation cell, the Berendsen thermostat and barostat are applied with the target $P-T$ conditions of 300 K and 0 GPa. We have performed NVT and NPT MD simulations and we found no significant variance in the results, indicating that there is no volume bias in the formation of A15 clusters and with respect to the other classes of clusters.

## Detection of A15 clusters in fcc structure

Identification of the A15 clusters in the structures from radiation damage calculations is performed using the distortion score of local atomic environments[63]. The task is solved in a feature space of atomic descriptors by distinguishing the learned atomic environments of A15 (inliers) from other structures (outliers) using Minimum Covariance Determinant (MCD)[64]. The atomic environments from the radiation damage calculations are compared to the learned fingerprints of A15 inclusions in fcc. Each atomic environment in the analysed system is characterised by a score (statistical distance), which describes its proximity to the learned A15 structures. This distance is compared to a decision threshold of the model and the atomic environment is classified as belonging to A15 if the distance is smaller than the critical threshold.

The computed distortion scores correspond to the robust statistical distance $d_{RB}$[63,64] from the centre of the training data cloud:

$$d_{RB}(\mathbf{x}_m) = \sqrt{(\mathbf{x}_m - \hat{\boldsymbol{\mu}}_0)^T \hat{\boldsymbol{\Sigma}}_{M_0}^{-1} (\mathbf{x}_m - \hat{\boldsymbol{\mu}}_0)^T} \tag{1}$$

where $\hat{\boldsymbol{\mu}}_0$ and $\hat{\boldsymbol{\Sigma}}_{M_0}$ are the MCD estimates of the data cloud centre and of the MCD statistical covariance, respectively[64]. Within the MCD formalism, the whole sample covariance matrix $\boldsymbol{\Sigma}_M$ is approximated by the covariance matrix $\boldsymbol{\Sigma}_{M_0}$ of a data subset with $M_0 < M$ points, for which the determinant of the sample covariance matrix is minimal[64].

The training data set for A15 detection consists of $I_{13}^{A15}$ clusters (2 connected complete and centred icosahedra built by 24 atoms) embedded in fcc structure with $a_0 = 4.045$ Å for Al and $a_0 = 3.615$ Å for Cu. In order to prevent the sensitivity of the model to atomic perturbations, a zero mean noise with normal distribution $\mathcal{N}(0, \sigma = 0.08$ Å$)$ was applied to the perfect atomic positions of the training structures. The model is trained on $M = 1840$ local atomic environments of A15 inclusions. The contamination factor is set to $\nu = 0.07$.

The training and test structural data for the MCD analysis is represented in the feature space of bispectrum b-SO(4) atomic descriptor[65] with the angular moment $j_{max} = 3.5$, resulting in 40 descriptor components per atom. For fcc Al and Cu, the cutoff distance is set to $R_c = 5.0$ Å and $R_c = 4.5$ Å, respectively.

## Calculations of A15 energy landscape and transformations

In this section we report efforts to characterise the energetic landscape of 3D A15 clusters and 2D interstitial dislocation loops governed by the same EAM potentials of Mendelev et al.[46] and Mishin et al.[47] that were used for radiation damage simulations. In order to investigate how the configurations of SIA clusters from different energetic basins transform one in another, it is necessary to go beyond standard molecular dynamics techniques, as the generated trajectories are too short to observe the thermally activated mechanisms by which point defect structures mutate and migrate. To investigate the energy landscape of interstitial clusters in Al and Cu, we have used two accelerated sampling approaches, ARTn[37] and TAMMBER[41].

The ARTn method[37] is an open-ended saddle search method composed of two main steps: the activation step and the relaxation step. The activation step consists in moving the system from a local minimum to a saddle point, achieved by following the lowest eigenmode of an approximate Hessian curvature matrix. The relaxation step consists in relaxing the system, from the computed saddle point to

another local minimum. Typically, computational efficiency requires the number of degrees of freedom to be restricted. In order to search for low activation barrier events starting from a particular state, we have searched over 400 events per ARTn simulation. We have used 5 series from each A15 cluster or dislocation loop basin. Events are accepted using a Metropolis algorithm with a temperature of 1000 K. The deformation around the defect is performed locally with a radius of 4.0 Å and partial Hessian is projected on a 15-dimensional Lanczos basis[37].

Exploring deeper into the energy landscape requires a method that can efficiently manage many search routines in parallel. The TAMMBER[41] code is a massively parallel sampling scheme that constantly deploys thousands of molecular dynamics workers, allowing a wide exploration of defect energy landscapes. When new minima are found, double-ended saddle search routines are used to find connecting saddle points to the existing network, whilst Bayesian analysis of the molecular dynamics data yields a robust measure of sampling completeness for each minima, namely an estimated rate to as-yet undiscovered regions of the energy landscape, represented by an absorbing sink. This information is used to build an off-lattice kinetic Monte Carlo, or Markov model, representation of the energy landscape, whose quality can be quantitatively measured by calculating the *residence time*: the expected trajectory duration before a previously unseen event occurs (time to absorption). Workers are then initialized according to a constantly updated distribution of starting states, such that the expected increase in residence time is maximal. TAMMBER is thus able to autonomously and optimally manage many thousands of workers, to rapidly construct an energy landscape of kinetically relevant configurations from an initial starting state, with minimal user supervision. Graph analysis on the atomic connectivity of minima is used to equate configurations reducible under translation, rotation, and identical exchange, to minimize duplication of effort. This affords massive efficiencies when studying point defects in crystalline materials[41,66].

To investigate the landscape of the $I_8$ cluster, TAMMBER was run for 6 h on 2000 cores, initialized with a single A15 starting configuration. The final energy landscape contained around 1000 states and over 2500 saddle connections, of which a representative subset is shown in Fig. 6c as a disconnectivity graph. The full landscape is provided in Supplementary Note 4.

The low energy system states identified by ARTn and TAMMBER were used as initial and final states for Nudged Elastic Band (NEB) calculations[38] in LAMMPS[62] in order to find a minimum energy path (MEP) between the two minima states (Fig. 6a, b, d). The MEP between these states is sampled with 24 points. The spring constant for parallel nudging force is set to 15.0 eV/Å.

## Calculations of diffuse scattering based on the data from DFT simulations

For the elastic scattering of some incident particles (e.g., X-rays or neutrons) by defect crystals, the scattered intensity for a momentum transfer $\mathbf{k}$ is given by the scattering function $S(\mathbf{k})$. The average of the scattering function should be taken over all possible microscopic defect configurations. The effect of thermal motion is very small compared to the defect signal. By subtracting the Bragg intensities from the scattering function, we obtain diffuse scattering, which for small concentrations of defects is proportional to both their concentration and average size; that is, $c\langle n \rangle$, where $c$ is the concentration and $\langle n \rangle$ is the average size of a defect cluster.

The experimentalists are interested in the diffuse scattering near the Bragg reflections, where they get particularly strong intensities. If the scattering vector $\mathbf{k}$ nearly coincides with a reciprocal lattice vector $\mathbf{g}$, we can write that $\mathbf{k} = \mathbf{g} + \mathbf{q}$, where $\mathbf{q}$ measures the deviation from the Bragg reflection and is usually assumed to be small compared to $\mathbf{g}$. We will denote the normalized direction of deviation by $\hat{\mathbf{q}} = \mathbf{q} / |\mathbf{q}|$, and similar notations will be adopted for other vectors. Tensors are written in boldface characters and their components in normal ones with the

corresponding indices. In the limit of small values of **q**, the scattering is determined by the long-range part of the displacement field, and it is possible to evaluate the results using elastic continuum theory. For crystals with a low density of point defects, the leading term for diffuse Huang scattering near Bragg reflections can be derived solely from the elastic constants of the material $C_{ijkl}$ and the defect dipole momentum $P_{ij}$ as follows:

$$S_H^{\mathbf{q}}(\mathbf{k}) = c\langle n\rangle|f_{\mathbf{g}}|^2|\mathbf{h}\cdot\mathbf{T}(\mathbf{q})|^2 \qquad (2)$$

Where $f_{\mathbf{g}}$ is the scattering factor, including Debye-Waller and polarisation factors, and **T** vector can be deduced from elastic constants and defect dipole momentum as follows:

$$T_i(\mathbf{q}) = \frac{1}{|\mathbf{q}|V}\sum_{j,l} G_{ij}\hat{q}_l\mathbf{P}_{jl} \qquad (3)$$

$$(\mathbf{G}^{-1})_{ij} = \sum_{k,l} C_{ijkl}q_k q_l \qquad (4)$$

The dipole tensor of the defect can be deduced directly from DFT calculations and the stress of the box with volume $V$, which is $P_{ij} = -V\sigma_{ij}$. In experiments, the average number of defects in a cluster is estimated from Eq. (2), which gives an intensity at a given **g** point in reciprocal space and for a given **q** direction[32,35]. Commonly, as in[31,32,35] and Fig. 3b, the morphology of defects can be identified from the ratio $S_H^{\mathbf{q},\perp}(\mathbf{k})/S_H^{\mathbf{q},\parallel}(\mathbf{k})$ of the Huang scattering in a direction perpendicular on **q** and other directions parallel to the same vector. The advantage of reporting the ratio is that the prefactors in Eq. (2) cancel out and the value is related solely to the magnitudes of the $P_{ij}$ tensor and elastic constants.

### Electron irradiation and defect characterisation

The discs of high purity Ni-0.4Ti (wt.%), with measured impurities C < 2, S < 4, O < 14, N < 1 in wt% ppm, of 3 mm diameter were mechanically polished to 70 µm thick, annealed at 1000 °C for 1 h, then electropolished for the Transmission Electron Microscope (TEM) observations. The in-situ irradiation was performed using the EM7 High Voltage Electron Microscope (HVEM) located at CEA-Saclay. Samples were irradiated by 1 MeV electrons at 450 °C. The electron flux was $4.5 \pm 0.1 \times 10^{18}$ e/cm$^2$/s or $5.4 \pm 0.1 \times 10^{-5}$ dpa/s considering a cross-section of 12 barns[67]. During the irradiation, a two-beam kinematic bright-field (KBF) mode was employed to image dislocations using a diffraction vector **g** = ⟨200⟩. Visible Frank loops and perfect loops can be distinguished by the stacking fault contrast inside Frank loops. The microstructure evolution was recorded in video by a high-speed camera. After irradiation, samples were post-characterised using a FEI TECNAI G2 TEM operated at 200 kV. Stereo-microscope and inside-outside methods[68,69] were applied to determine the nature of dislocation loops (see Supplementary Note 7 for more details).

## Data availability

The data presented in this paper are available at the request of the corresponding authors.

## Code availability

The MILADY package is open-source software under ASL license and can be downloaded at https://ai-atoms.github.io/milady/. The TAMM-BER code can be downloaded at https://github.com/tomswinburne/tammber.git.

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

## Acknowledgements

This work was financially supported by the Cross-Disciplinary Program on Numerical Simulation of CEA, the Cross-cutting basic research Program of the Energy Division of CEA (RTA Program), the RMATE project of CEA, the French Alternative Energies and Atomic Energy Commission, and the NEEDS program (CNRS-CEA-EDF-ANDRA-AREVA-IRSN-BRGM). A.M.G. and M.C.M. acknowledge the support from GENCI - (CINES/CCRT) computer centre under Grant No. A0130906973. C.D. acknowledges the support from GENCI - CINES computer center under Grant No. A0050910624. C.D. and M.C.M.

acknowledge PRACE support from MARCONI-KNL at CINECA under Grant No 2016153636. A.C. acknowledges the access to the HPC resources of the TGCC computing centre, under the DARI allocation A0100911528. T.D.S. gratefully recognizes support from the Agence Nationale de Recherche (ANR), via the MEMOPAS project ANR-19-CE46-0006-1, and IDRIS resources under DARI allocation A0090910965. This work has been carried out within the framework of the EUROfusion Consortium, funded by the European Union via the Euratom Research and Training Programme (EUROfusion Grant No. 101052200). The views and opinions expressed herein do not necessarily reflect those of the European Commission. M.C.M. and A.M.G. gratefully thank Dr. Emmanuel Clouet for many insightful private conversations concerning dislocation interactions; and Dr. Maylise Nastar for the interesting discussions regarding the formation of precipitates. The electron irradiation experiments were supported by the French EMIR&A accelerators network. K.M. and M.L.P. are grateful for the support from Dr. Olivier Tissot and Thierry Vandenberghe on the operation of the High Voltage Electron Microscope.

## Author contributions

A.M.G. and M.C.M. designed the study. C.D., M.C.M., and A.M.G. performed DFT calculations. A.C. performed the FPA simulations. C.D. carried out the simulations of displacement cascades. A.M.G., T.D.S., A.D., J.C., and M.C.M. performed the exploration of energy landscapes. A.M.G. performed the analysis of all simulations and structures. A.M.G. and M.C.M. designed the database and fitted the ML potentials for Ni. M.L.P. and K.M. conceived and conducted the in-situ TEM experiments and performed characterisation and interpretation of the observed microstructures. A.M.G. and M.C.M. wrote the manuscript. All authors discussed the results and commented on the manuscript.

## Competing interests

The authors declare no competing interests.
