## [Peer Review File · Nature Communications]

Compact A15 Frank-Kasper nano-phases at the origin of dislocation loops in face-centred cubic metalsReviewers' comments:

Reviewer #1 (Remarks to the Author):

Referee's report on the manuscript 'Compact A15 Frank-Kasper nano-phases as the primary source of dislocation loops in face-centred cubic metals', by Goryaeva et al. submitted to Nature Communications.

This manuscript describes the results on DFT and classical MD modeling of A15 and planar interstitial clusters in Al and Cu. The conclusion made that small interstitial clusters are stable as 3D A15-type structures that can be transformed into planar perfect or Frank loops when growing. It is discussed that A15 structures are sources of dislocation loops, and this has an impact on microstructure evolution under irradiation.

This research is in line with the earlier studies on C15 structures in bcc metals and the subject is interesting for the radiation damage community. While the simulation results are interesting, the reviewer has certain doubts on the suitability of this for publishing as a Nature Communication article. The main concern is about the importance and reliability of the presented results. The existence of small 3D A15 sessile interstitial clusters is unlikely to have a significant impact to the microstructure evolution under irradiation. At least, the authors did not present enough evidences of the possible impact.

Below are my more detailed comments:

First, it seems to me that the title is a bit confusing. Reading 'compact A15 Frank-Kasper nano-phases as the primary source of dislocation loops' one may expect to see a real source of dislocation loops, but in fact this is just a transformation of an interstitial clusters in A15-phase into either sessile or glissile dislocation loop. I am not sure that this can be called a source.

Second, while all modelings with empirical potentials look quite reasonable, they can be taken as qualitative only, because of the uncertainty with empirical potentials. One may assume that the potentials used reproduce the *transformation mechanisms*, but very unlikely they reproduce the exact *critical size* of A15 clusters. Thus, DFT calculations predict that *some* of A15 clusters may be more favorable than 2D arrangements having less than ~4-8 interstitial atoms, whereas EAM potentials, e.g. in Al, expand this for $\frac{1}{2}\langle 110 \rangle$ clusters up to >20 interstitials. In these estimations I trust more to DFT results.

Third, DFT calculations were made at zero T, that means dynamic stability of these clusters is unknown. Therefore, one cannot conclude that they really can be formed directly in cascades or/and grow due to dynamic interactions between single mobile $\langle 100 \rangle$ dumbbells.

Fours, even if one accepts that A15 clusters, say up to 10 interstitials, are stable, what is the consequence of this for the microstructure evolution under irradiation? I cannot see any qualitative changes, and the authors did not present any quantitative estimations. For example, what is the

difference for microstructure evolution if these clusters are A15 or sessile Frank loops?

And finally, extensive production and accumulation of sessile clusters of several interstitial atoms would be reflected in experimental studies. For example, recovery of electrical resistivity should observe the stage/s related to instability of these clusters. Interstitial defects contribute significantly to the resistivity increase so such a phenomenon could be noticed.

Concluding, I would like to repeat my above statement: this work reports an interesting idea and results. However, the significance of the presented results is not well justified for the high standards of Nature Communication publication.

To my opinion the manuscript could be considered for publication if the authors a) prove that A15 clusters are dynamically stable at the large enough number of interstitials and b) justify the importance and reliability of their results for the microstructure evolution.

Reviewer #2 (Remarks to the Author):

Compact A15 Frank-Kasper nano-phases as the primary source of dislocation loops in face-centred cubic metals

Goryaeva et al

In this manuscript, the authors use their newly developed distortion score method to characterize very small interstitial defect clusters containing fewer than 10 point defects, and find that rather than being best described as collections of $\langle 111 \rangle$ or $\langle 110 \rangle$ dumbbells (which would be consistent with calling them small loops), they are better described as collections of $\langle 100 \rangle$ dumbbells, consistent with A15 phase. This view is supported by electronic structure calculations which find very similar energies for the different clusters, and Frenkel pair accumulation simulations which show the appearance of A15 clusters.

My biggest question to the authors, given that it has been submitted to Nature, is why this is an important distinction at such small sizes. The authors themselves acknowledge that it would be difficult to say whether black dots in TEM micrographs should be one cluster type or another, and that the energy barrier for dumbbell rotation or cluster transformation is order 0.5 eV, so is active at room temperature while inactive in their simulations.

If the clusters were to grow large, as has been observed for C15 clusters in bcc iron, their sessile nature could well impact hardness. But at 7-8 interstitials is there any effect on material properties expected?

If there were a way to experimentally determine the nature of these clusters, or their effect, this paper would clearly be worthy of Nature Comm. As it is, I have some technical questions below which might

help the authors strengthen the theoretical argument made.

Figures 3 and 4:

I wanted to find the relative importance of small A15 vs small loops by comparing the proportion of interstitials in each category, as line 190 states

"The number of A15 clusters detected in this work per (MD) cascade is a small and represents only 4-5 % of interstitial atoms."

while fig 4 suggests the number of A15 clusters decreases with dose (to 0.01dpa), and line 260 reads "No mixed structures of A15 with dislocation network were observed in our simulations"

If I am reading it correctly, figure 3 shows the number of clusters in each category, not the number of interstitial atoms in each category. Figure 4 could give the number of interstitials in A15, but not the number in the loops (we can't necessarily convert a line length into an area).

Could the authors provide a simple comparison for the number of interstitials in each category, please?

In figure 4 the histogram seems to start with A15 clusters of size 2. Is this a 100 dumbbell plus the octahedral interstitial? This seems very high energy- surely it would symmetrise to a triangular structure? Can the structure be shown in figure 2, please?

In figure 4 the x-ranges seem to be different for A15 and loops. Is there a good reason, or can they be made the same?

In figure 4a) the small A15 clusters are shown. I can't really see how they relate to the clusters shown in figure 1.

Line 126:

"The distribution of A15 cluster sizes in Al have the shape of Poisson-like distribution"

I didn't really understand why this should be so, and frankly can't see it myself in figure 4b. (I tried to extract a couple of size-frequency histograms, they did not look particularly Poisson to me.)

The authors should clarify why they think the probability of an injected interstitial joining a cluster is independent of the cluster size in the FPA method.

If this can be done, it would be a useful contribution as it simplifies hugely the size-frequency histogram.

On a related note: line 152 states "the distribution of A15 cluster size [for Cu] does not follow a Poisson-like distribution"

Was a chi-squared test done? Is this assessment carried out by eye?

3D vs 2D interstitial clusters.

In line 247 we read "A complete A15 icosahedron is built by 7 interstitial atoms", and in line 259 "Due to the important concentration of pressure within the cluster, A15 fully transform into loops"

But a 7 interstitial loop also has 7 interstitials, so should have a comparable relaxation volume. If the authors want to make the claim that there is an elastic relaxation made possible by the transformation from A15 to loop, they should provide the dipole tensors or relaxation volumes.

On a related note:

If there is a significantly different dipole tensor for the A15 and loop structures, then they should become preferentially formed in different strain states.

How did the volume of the simulation cell change during the defect accumulation? Is it sufficient to tip the balance in favour of one or other structure?

Could the uniaxial strain expected during ion irradiation or a TEM foil experiment lead to a different conclusion about the balance?

Line 259:

"A15 fully transform into loops. No mixed structures of A15 with dislocation network were observed in our simulations"

Is this consistent with figures 3c),3d), and 4b)? There seem to be plenty of A15 structures in these figures.

Line 284:

"These observations [of 10dpa Al at 400K] are in agreement with our DFT and FPA simulations"

Consistent perhaps, but not really agreement. The FPA simulations are 0.01dpa at very low temperature.

Line 418:

I am concerned by the statement

"atoms are randomly displaced in such a way that their distance from any other atoms is not smaller than 1.4 Å."

I understand that this has computational benefits, in that a possible singularity in the potential energy surface is avoided, and large forces are not generated which could perhaps start a displacement cascade.

But surely this cutoff also introduces biases.

Firstly, the higher the cutoff length, the more it will tend to make interstitial insertion preferentially into open volumes, and so into lower energy positions.

This could have the effect of promoting the growth of one type of interstitial cluster over another. As this is the main theme of the paper, this possibility should not be lightly ignored.

Secondly, as the cutoff is apparently the same for both Cu and Al, elements with rather different lattice parameters, the effect of selecting one interstitial position over another for Cu and Al changes, perhaps giving a different or exaggerated conclusion.

As a (very rough) estimate, I find that the ratio of acceptable volumes for tetra:octa positions in pristine fcc lattice is 2.5:1 for Cu and 1:1 for Al.

I can't say if this will distort the findings, but a convergence study should be performed to confirm it does not have an effect.

Detection of A15 clusters in fcc

As the classification of clusters is vital to this work, it is important that it is robust.

My concern is that a very small cluster of 3-4 interstitials, which people have previously not really known how to correctly label, is being called A15 structure with high confidence here.

From what I have seen, I believe the authors' new "distortion score" method is a good way of discriminating structures.

The question mark I have comes from line 445: "its proximity to the learned A15 structures", which might suggest that the algorithm has been told to classify small clusters as A15.

Was a proximity to "learned loop structures" considered?

I note in line 133 "many interstitial Frank loops appear at the size of 7-8 interstitial atoms".

Is it possible for a 3-4 interstitial cluster to be labelled a Frank loop?

Were any 3-4 interstitial clusters labelled anything other than A15?

I see "noise with normal distribution $N(0;0.08\text{Å})$ was applied on the perfect atomic positions", which

seems fine for Cu at 300K. But Al has a larger Debye-Waller factor. Should the same value be used? Does it matter?

What account is made for local strain? Do the possibly large local strains have an impact for these dense microstructures?

On a related note for figure 4:

How was it determined if an interstitial cluster $N < 10$ was Frank or Prismatic?

How was the loop perimeter determined for small clusters < 10 interstitials? Ovito's DXA method generally fails at small size because there is no Burgers circuit through perfect material.

The authors should consider how to make the argument compelling that the symmetry of the small clusters is that of $\langle 100 \rangle$ dumbbells not $\langle 110 \rangle$ or $\langle 111 \rangle$, and how they define loops $N < 10$.

As a general note, there are too many grammatical errors and spelling mistakes in the manuscript.

Poorly proof-read lines like 128 or 432 should not make it to referees:

"This distribution is not biased by attractive configuration and sample a pure metastable basin"

"... with a transition to the equilibrium potential between 1.7 to the equilibrium potential between 1.7 and 2.0A and 2.0A."

Reviewer #3 (Remarks to the Author):

This manuscript presents interesting computational results that suggest the initial stages of point defect clustering in FCC metals may be more complex than historically assumed. Overall, the paper is well written and the authors use appropriate caution regarding the general validity of the results. I have a few minor comments.

1. The authors should consider some alternative wording in lines 37-40 (p. 2). In line 37, the statement "defects... aggregate in voids that form SFTs" implies that 3D voids are first formed and then transform into SFTs, which is contrary to the mainstream understanding of vacancy cluster evolution; in general, it is extremely rare for void-SFT transformation (or vice versa) to occur, due to the enormous local atomic rearrangement that is required for vacancy cluster sizes exceeding ~ 10 defects. Line 38 implies that the vacancies invariably first cluster into a 3D void before collapsing into a planar loop; this is contrary to the current conventional thinking where loops are formed via one-by-one addition of vacancies, as noted elsewhere in the manuscript. In addition to the modeling studies that have predicted SFT formation from vacancy loops (lines 38-39), there are numerous in-situ TEM studies (e.g., Matsukawa, Science vol. 318 (2007) p.959, etc.) that have observed vacancy loop conversion to SFT. Reports claiming conversion of SFTs to voids (refs. 15, 19] discussed on p. 2, line 37 of the manuscript need further confirmation (there are far more numerous experimental and computational studies that have not observed such a

conversion).

2. The introduction section discusses both vacancy and interstitial defect clusters, whereas the modeling results only examine interstitial cluster evolution. Is the A15 cluster mechanism considered to potentially also occur in vacancy cluster evolution?

3. The author definition for the term “loop size” should be indicated in the text; does this mean radius, diameter, or loop circumference? On p. 8, line 132 the manuscript states that a loop size of 2.0 to 2.1 nm corresponds to 7-8 atoms, which seems nonphysical if “size” means radius or diameter. Please recheck.

4. On p. 19 (Methods), the potential for artifact(s) associated with the extremely high damage rate ($\sim 3 \times 10^5$ dpa/s) for the FPA simulations that will introduce extremely high defect supersaturations should be briefly noted.

5. In the methods section (discussion of Eq. 1) or elsewhere (introduction?), in addition to the Hirth and Lothe reference the authors should consider citing one of the numerous journal articles that have calculated the relative energies of perfect and faulted loops, e.g., Kroupa Czech J. Phys B 1960, Zinkle et al. Phil Mag A 1987, or Povstenko J. Phys. D 1995.

5. Several minor typos should be corrected in the manuscript.

p. 2, line 49: “other” -> “others”

p. 3, line 71: “octahedral” -> “octahedral”

Fig. 3 caption (3rd line): “Microstrucure” -> “Microstructure”

p. 7, line 125: duplicate word “over over” -> “over”

p. 12, line 228: “dislocations” -> “dislocation”

p. 13, line 249: “then” -> “than”

Response to reviewers

Manuscript Ref.: NCOMMS-21-39664A-Z

Compact A15 Frank-Kasper nano-phases at the origin of dislocation loops
in face-centred cubic metals

February 16, 2023

We thank the Referees for the thorough review of our manuscript and the recognition of our work as important for the communities of physics and materials science. Their thoughtful comments enabled us to bolster our argumentation and amplify the paper's impact, making it more appropriate for the wide readership of Nature Communications.

In order to improve the quality of the paper and to answer the constructive criticism of the Referees, a significant revision of the manuscript has been performed. In particular, the fundamental changes include:

- solid argumentation of A15 cluster formation based on experimental observations. We are sincerely grateful to Referee 1 for his/her valuable advice to carefully examine the literature regarding the resistivity recovery experiments that are relevant to the current discovery;
- in light of available experimental data, we provide additional DFT simulations of A15 clusters and dislocation loops in fcc Ni;
- a new section discussing the importance of A15 cluster formation in fcc metals for microstructural evolution is introduced, following the recommendations of Referee 1 and Referee 2;
- *in-situ* electron irradiation of Ni-0.4Ti is performed to demonstrate how A15 clusters can account for the development of diverse loop morphologies, which can not be explained by the conventional mechanism;
- modification of the title.

In the revised version of the manuscript, the changes appear in green.

We believe that the present corrections address the main concerns expressed by the Referees and significantly improve the manuscript. We hope that the applied corrections together with the important extension of Supplementary Materials will lead to the acceptance of the paper.

Below we provide our point-by-point replies to the Referees' comments.

Reviewer #1

This manuscript describes the results on DFT and classical MD modeling of A15 and planar interstitial clusters in Al and Cu. The conclusion made that small interstitial clusters are stable as 3D A15-type structures that can be transformed into planar perfect or Frank loops when growing. It is discussed that A15 structures are sources of dislocation loops, and this has an impact on microstructure evolution under irradiation.

This research is in line with the earlier studies on C15 structures in bcc metals and the subject is interesting for the radiation damage community. While the simulation results are interesting, the reviewer

has certain doubts on the suitability of this for publishing as a Nature Communication article. The main concern is about the importance and reliability of the presented results. The existence of small 3D A15 sessile interstitial clusters is unlikely to have a significant impact to the microstructure evolution under irradiation. At least, the authors did not present enough evidences of the possible impact.

Below are my more detailed comments:

We thank the Referee for the attentive reading of our manuscript. In particular, we acknowledge the constructive remarks indicating the lack of convincing arguments that emphasize the importance of A15 clusters.

This paper draws the attention of the broad audience of Nature Communications to the fact that the fundamental mechanisms of interstitial defect formation are more complex than historically assumed and require a general revision.

Therefore, we argue that this study is not simply a follow-up of C15 cluster studies. Previously, the formation of three-dimensional SIA clusters has been considered by the community as a particular case, relevant exclusively for body-centered cubic iron and commonly related to its magnetism [1, 2, 3]. In this work, we demonstrate that the formation of compact 3D clusters prior to dislocation loops is rather a general phenomenon. The formation of such clusters has been systematically disregarded by the materials science community for the last sixty years.

Below we provide our point-by-point answers to the issues raised by the Referee. We believe that our replies together with the significant changes in the manuscript (including experimental evidence and additional theoretical predictions) provide the necessary arguments.

R1-1. First, it seems to me that the title is a bit confusing. Reading ‘compact A15 Frank-Kasper nano-phases as the primary source of dislocation loops’ one may expect to see a real source of dislocation loops, but in fact this is just a transformation of an interstitial clusters in A15-phase into either sessile or glissile dislocation loop. I am not sure that this can be called a source.

We thank the Referee for this comment. Our work demonstrates that A15 clusters are at the origin of interstitial dislocation loops in fcc materials, despite the common belief that such loops form directly through the diffusion and gradual accumulation of SIA dumbbells, without any intermediate stages. The term “source” has been used in the title with the meaning of “origin”. If we understand correctly, the Referee suggests that the term “source” can be confusing, being too close to the terminology related, for instance, to Frank-Read dislocation sources.

- We have changed the title of the paper to **“Compact A15 Frank-Kasper nano-phases at the origin of dislocation loops in face-centered cubic metals”**

R1-2. Second, while all modelings with empirical potentials look quite reasonable, they can be taken as qualitative only, because of the uncertainty with empirical potentials. One may assume that the potentials used reproduce the transformation mechanisms, but very unlikely they reproduce the exact critical size of A15 clusters. Thus, DFT calculations predict that some of A15 clusters may be more favorable than 2D arrangements having less than 4-8 interstitial atoms, whereas EAM potentials, *e.g.* in Al, expand this for $\frac{1}{2}\langle 110 \rangle$ clusters up to > 20 interstitials. In these estimations I trust more to DFT results.

We thank the Referee for this remark. There is no doubt that DFT is more reliable than any EAM potential. We apologize for not being explicit enough on this point in the paper.

We employ EAM potentials that predict the same ground states as DFT. Both DFT and EAM suggest a competition between the A15 clusters and *the most stable* loop family up to 7-8 SIAs, *i.e.* there is a good *qualitative agreement*. The remark of the Referee is most likely related to Figure 2d, where A15 clusters in Al are shown to be more stable than $\frac{1}{2}\langle 110 \rangle$ loops up to 20 SIAs. However, this prediction does not concern the ground state of dislocation loops in Al. DFT and EAM consistently suggest that $\frac{1}{2}\langle 110 \rangle$ family is less stable than $\frac{1}{3}\langle 111 \rangle$ loops. Thus, the pertinent information on the relative stability of A15 clusters with respect to the most stable 2D family, *i.e.* $\frac{1}{3}\langle 111 \rangle$ loops, was presented in Fig. 2c. This important stability is qualitatively well described by the EAM potential.

We admit that our representation of the defect stabilities was not very clear and did not emphasize the relative stability of 2D and 3D clusters. For the sake of clarity, we have performed the following changes.

- **Figure 2** has been modified in order to accentuate the relevant information. The revised picture displays only DFT results and the histograms show the relative stability of A15 clusters vs the *most stable dislocation loop* family. The indication of the defect types is now clearly provided in each histogram plot and in the figure capture.
- The detailed comparison of formation energies and relative stability predicted by DFT simulations and semi-empirical potentials for different cluster types is now provided in **Supplementary Note 1**. The references to this comparison are clearly provided in the relevant parts of the main text.
- We compare the critical size of 3D clusters predicted by our DFT simulations to those from experiments [4] alone in order to demonstrate that DFT simulations are given more importance than semi-empirical potentials (see **Figure 3** in the revised manuscript and our answer to **R1-5** for more details).

R1-3. Third, DFT calculations were made at zero T , that means dynamic stability of these clusters is unknown. Therefore, one cannot conclude that they really can be formed directly in cascades or/and grow due to dynamic interactions between single mobile $\langle 100 \rangle$ dumbbells.

We thank the Referee for this comment. This point was probably not emphasized very well in the paper. Indeed, we compute the formation energies with DFT at 0 K and compare them with the corresponding energies obtained with EAM potentials. Then, we perform the FPA simulations using the validated EAM potentials and observe there the formation of A15 clusters both in Al and Cu. Here it is worth noting that, in the case of Al, 0 K relative stability of A15 clusters vs dislocation loops is in good qualitative agreement with DFT, whereas, for small-size clusters in Cu, the EAM suggests that A15 is less stable compared to DFT. Thus, in fcc Cu, one can rather expect the formation of dislocation loops instead of A15 clusters. However, even with the potential predicting A15 as less stable, these compact 3D clusters do form, as reported in Figure 3b,d (of the submitted version of the manuscript). Interestingly, a similar observation has been recently made for the formation of compact C15 clusters in bcc W [5] where they are expected to be less stable than 2D dislocation loops.

In the revised manuscript, we provide experimental evidence that our theoretical predictions of A15 clusters are valid for real-world systems (see the answer to **R1-5** for more details). In **Figure 3** of the revised manuscript we compare our 0 K DFT results with experimental findings at finite temperature during the defects nucleation in resistivity recovery experiments [4]. There is a remarkable agreement between simulations and experiments for Al, Cu, and satisfactory agreement for Ni. Further investigations of the finite-temperature effects on the defect formation mechanism is an excellent perspective for future studies, however, it is far beyond the scope of the present paper.

R1-4. Fours, even if one accepts that A15 clusters, say up to 10 interstitials, are stable, what is the consequence of this for the microstructure evolution under irradiation? I cannot see any qualitative changes, and the authors did not present any quantitative estimations. For example, what is the difference for microstructure evolution if these clusters are A15 or sessile Frank loops?

We thank the Referee for this question. We agree that a discussion regarding the possible effects of A15 clusters on microstructural evolution was missing in the initial version of the manuscript. Moreover, this discussion became imperative in light of new theoretical predictions in Ni and the experimental evidence that we provide in the revised manuscript. The latter is based on the Referee's recommendation in **R1-5**, which we gratefully acknowledge.

We believe that the mechanism of dislocation loop formation via A15 clusters cannot be neglected even if the transition A15 clusters \rightarrow dislocation loops occurs at a relatively small size. Below we list three major topics, which should be carefully investigated in future theoretical and experimental studies.

Firstly, the emergence of finite-size dislocation loops from dense nano-phase clusters radically changes our vision of the mechanisms that influence microstructural evolution. Until now, only a continuous mechanism of loop formation without any intermediate stages was considered. Although the size of A15 clusters is not big, we argue that the existence of this intermediate step in dislocation loop formation is a paramount point with ample consequences. The materials science community should revise its multiscale models and quantify the impact of the present findings. Such developments are worth further thorough studies. However, performing such studies to provide quantitative estimations is far beyond the scope of the present paper. We aim to disclose the existence of dense clusters that form prior to dislocation loops and provide a discussion on the possible qualitative impact on microstructure in order to motivate the scientific community to critically reconsider the fundamental mechanisms of dislocation loop formation.

Secondly, not only the size of the A15 clusters is important but also the density of clusters in the matrix. The evolution scenarios with or without immobile 3D clusters are very different and somewhat not very sensitive to the size of 3D clusters [6, 7] (which generally remains below the experimental detection limit). However, their existence, which we point out in this study, can change our knowledge and understanding of microstructural evolution. Quantitatively estimating this problem is a beautiful prospect for this study. For example, let's consider the case of immobile C15 clusters in bcc metals, which appear before dislocation loops [1, 2, 8]. The characterization of microstructural evolution of bcc Fe and W obtained with multiscale techniques such as kinetic Monte-Carlo or cluster dynamics [9, 6, 5, 7] suggests that the presence of immobile C15 clusters drastically changes the microstructure. In particular, two very recent studies [6, 7] clearly demonstrate that accounting for the C15 Laves phase clusters in multiscale models leads to an increase in the interstitial density at low doses compared to the models without C15 clusters. These results are in better agreement with experiments. Thus, further investigations of dense clusters like A15 and their impact on the microstructural processes represent an outstanding perspective for the future studies

Thirdly, the mechanism of loop formation can bias their morphologies in materials under stress or in a strain field, *e.g.* in thin foils or in the presence of a surface. The dipole tensor of A15 clusters has cubic symmetry (see the table in the reply to the question R2B-2a of the second Referee or Supplementary Note 6). Thus, unlike that of dislocation loops [10, 11, 12, 13, 14], it has no directional preferences. This means that during the transformation from A15 cluster \rightarrow dislocation loop, elastic interactions with the strain field will induce certain preferential habit planes of dislocation loops, which will correspond to the lowest elastic interaction energy (for example, elastic energy can be used as in Eq. 28 of the review paper [14] or the treatment in the general book for elasticity [15]). Consequently, the mechanism of A15 cluster \rightarrow dislocation loop transformation can impact the morphology of dislocation loops under strain.

- In order to address the point raised by the Referee and better emphasize the role of A15 clusters in the microstructural evolution, we have added a new section to the discussion of the paper entitled: **On the role of A15 clusters in microstructural evolution**, which describes and qualitatively compares the two loop formation scenarios: via A15 clusters and by the direct accumulation of SIA dumbbells into 2D clusters.

R1-5. And finally, extensive production and accumulation of sessile clusters of several interstitial atoms would be reflected in experimental studies. For example, recovery of electrical resistivity should observe the stage/s related to instability of these clusters. Interstitial defects contribute significantly to the resistivity increase so such a phenomenon could be noticed.

We gratefully acknowledge the Referee for this remark. We agree that in the originally submitted manuscript, we were mainly focused on solid theoretical proofs, but we have not provided enough argumentation based on the experiments. As suggested by the Referee, a comparison of our simulations with the observations from resistivity recovery experiments is an excellent case to validate our theoretical predictions and to strengthen the message of the paper. In order to reinforce the argumentation, we have added a section that compares our theoretical predictions with the Stage II resistivity recovery experiments in Al, Cu, and Ni.

In Figure 1a below and **Figure 3** of the manuscript, we summarize the results of experimental studies that

combine resistivity recovery experiments and diffuse X-ray scattering in Al, Cu, and Ni from the review book by Ehrhart *et al.* [4] and compare them with our theoretical predictions from DFT calculations. For all considered metals, there is a common trend in the formation of small SIA clusters. At the end of Stage I, small clusters of 2-4 SIAs are formed. Then, during Stage II, larger clusters are observed. Diffuse X-ray scattering indicates that these clusters have a 3D structure with up to 6-7 SIAs in Al and Cu, and 20-30 in Ni. For bigger cluster sizes, the same experimental method indicates the appearance of dislocation loops. The experimentally observed crossover size in Al and Cu are in very good agreement with our theoretical predictions based on the 0 K DFT energy landscape (Fig. 1a). For Ni, the theoretical simulations suggest a crossover of 10-12 SIAs, while the experiment results provide much larger crossover values between twenty and thirty SIAs. We can not provide a straightforward explanation of this discrepancy, however, one can still indicate at least two reasons, described below. (i) The difference in the formation energy of the A15 family and $\frac{1}{3}\langle 111 \rangle$ and $\frac{1}{2}\langle 110 \rangle$ is not sufficient to indicate the transition size in Ni. The formation energies in Ni are nearly twice bigger than those in Al. One can also expect the transformation barrier from A15 to a loop to be higher. When applying a factor of two (from the formation energies) to the migration barrier between A15 and loops compared to Al and Cu, we obtain athermal barriers with more than one eV that can trap the SIAs in attraction basins of A15 even for sizes much larger than 10 SIAs. (ii) The experimental value of the A15 \rightarrow loops crossover was determined at 300 K. Possibly at this temperature, the vibrational and magnetic entropy can stabilize the A15 nano-phase clusters up to larger sizes than 10 SIAs. Both of these points are suppositions, which open up many perspectives for future exhaustive and quantitative studies.

Further, we consider in detail the case of Ni of the experimental study of Bender *et al.* [19] that provides experimental values of average defect size by recording simultaneously the resistivity recovery and diffuse Huang scattering experiments. Having the accurate values of the defect dipole tensor P_{ij} from DFT calculations of A15 and dislocation loops up to 25 SIAs, we are able to reinterpret those experiments. The measurements in Ref. [19] were performed over the first three stages of fcc Ni irradiated with electrons: Stage I below 70 K, Stage II between 70 K and 300 K, and Stage III between 300 K and 500 K. Within the first stage, $I_{D,E}$ SIAs begin to agglomerate, and at the end of Stage I, small clusters of $\langle n \rangle = 2-3$ interstitials are formed. During Stage II, these clusters slowly grow at $70 \text{ K} < T < 200 \text{ K}$, and an important increase in cluster size is observed between 200 K and 300 K. Here is the exact citation from Ref. [21]: “...the rapid increase of $\langle n \rangle$ is accompanied by a change in the defect structure of the agglomerates, i.e. a rearrangement of three-dimensional interstitial agglomerates into interstitial loops on $\{111\}$ lattice planes ... This rearrangement occurs for $\langle n \rangle$ between 20 and 30 interstitial atoms.”

The size of the clusters is estimated by the intensity of Huang scattering and the resistivity values at a given temperature. The geometry of defects (3D versus 2D) is deduced by the characteristic values of scattering function $rS_H = S_H^\perp/S_H^\parallel$ (as in Figure 1b reproduced from Ref. [19]). This function rS_H can be computed for each defect from the elastic constants of the material and the accurate values of the elastic dipole tensor of the defect (see Methods of the revised manuscript, page 21). Figure 1b reports the values of rS_H function for A15 and $\frac{1}{3}\langle 111 \rangle$ loops. Using the formalism described in Methods (revised manuscript, page 21), we compute $rS_H = S_H^\perp/S_H^\parallel$ from *ab initio* values of P_{ij} of defects. We have used the defect dipole tensors P_{ij} from 4 SIAs up to a size of 24 SIAs, from our DFT calculations. For each type of defect, we compute rS_H function at the conditions given by the experiment [21] (*i.e.* $\mathbf{q} = [010]$ direction at the point of reciprocal lattice $\mathbf{g} = (400)$ using the convention described in **Methods: Calculations of diffuse scattering based on the data from DFT simulations**). This theoretical prediction can be directly compared with experimental findings, as reported below in Figure 1b. The computed values of rS_H for A15 clusters are close to zero, while for $\frac{1}{3}\langle 111 \rangle$ they range between 40 % and 60 % (in general proportional to the size). The values associated to perfect loop $\frac{1}{2}\langle 110 \rangle$ are much larger, around 200 %. The most important aspect is that there is a gap in values of rS_H between A15 and $\frac{1}{3}\langle 111 \rangle$ loops. There are no intermediate values between 0 % and 40 %, regardless of the SIAs cluster size. The Huang scattering signal is proportional to the density of various morphologies of SIAs (see the section **Calculations of diffuse scattering based on the data from DFT simulations** in Methods). The intermediate experimental values around 20-30 % of rS_H come from the average between the almost zero signal of A15 3D clusters and the signal of dislocation loops, which start to form from A15 clusters in

Figure 1: **Growth of interstitial clusters in fcc metals.** (a) The experimental findings in diffuse X-ray scattering and resistivity recovery experiments in Al [16, 17], Cu [18] and Ni [19, 20, 18] compared with the theoretical predictions of this work. The average size of clusters (in the number of SIAs) determined in experiments is reported as a function of temperature, while our predictions are based on 0 K DFT simulations. The curves for Al, Cu, and Ni are shown in yellow, green, and purple, respectively. For each metal, the interval of Stage II is indicated with a similar color. The experimental data are taken from Ref. [4]. The open circles indicate the experimental size-temperature estimation of the dislocation loop formation. Red rectangles report the critical size of A15 clusters from our DFT simulations. (b) The experimental measurement in Ni, from Ref.[19], of the ratio of $S_H^{\perp}(\mathbf{k})/S_H^{\parallel}(\mathbf{k})$ (see Methods) of the Huang scattering in the directions perpendicular and parallel to \mathbf{q} . The colored rectangles in purple - for A15 nano-phase clusters - and green - for $\frac{1}{3}\langle 111 \rangle$ Frank loops - are the present theoretical estimation of the range of the previously defined scattering ratio from the DFT values of elastic dipole tensor P_{ij} of various defect morphologies having sizes between 2 and 25 SIAs clusters. (c) Schematic illustration of interstitial-type defects stability in fcc Al, Cu, and Ni. The cluster size N is the number of SIAs. The dashed areas indicate the size of clusters for which metastability phenomena can manifest, giving rise to the coexistence of at least two defect types. (d) TEM micrographs of *in-situ* electron irradiation of Ni-0.4Ti recorded under two-beam Kinetics Bright Field conditions using $g = \langle 200 \rangle$.

the end of Stage II. Thus, Figure 1b suggests that for small SIA clusters, there are no defects other than A15 clusters, as long as the rS_H signal is close to zero. The above results unambiguously demonstrate that the 3D defects predicted by the experiments are clusters that are energetically stable and kinetically trapped in the attraction basin of the Frank-Kasper A15 nano-phase with an almost zero rS_H signal.

We sincerely acknowledge the Referee for the suggestion to have a better look at the resistivity recovery experiments. The above astonishing finding provides solid proof of A15 nano-phase formation in fcc matrix. Direct comparison with experimental observations has considerably improved the quality of the paper. To emphasize our findings, the following changes to the manuscript have been done.

- A new section entitled: **Evidence of the A15 clusters formation from experiments** is added in the Results of the main manuscript.
- New **Figure 3** compares theoretical predictions with experiments.
- The **Discussion** section has been revised taking advantage of the new findings.
- A new section entitled: **Calculations of diffuse scattering based on the data from DFT simulations**, is added in the **Methods**.

R1-6. Concluding, I would like to repeat my above statement: this work reports an interesting idea and results. However, the significance of the presented results is not well justified for the high standards of Nature Communication publication. To my opinion the manuscript could be considered for publication if the authors a) prove that A15 clusters are dynamically stable at the large enough number of interstitials and b) justify the importance and reliability of their results for the microstructure evolution.

We express our deep gratitude to the Referee for the valuable advice to search for supporting evidence in the literature of resistivity recovery experiments related to the subject of the present paper. We believe that our experimental and theoretical arguments provide sufficient evidence of A15 nanophase formation in fcc metals and clarify their role in microstructural evolution. We hope that our replies to the Referee's query (**R1-4** ad **R1-5.**) will lead to the acceptance of the manuscript.

Reviewer #2

In this manuscript, the authors use their newly developed distortion score method to characterize very small interstitial defect clusters containing fewer than 10 point defects, and find that rather than being best described as collections of $\langle 111 \rangle$ or $\langle 110 \rangle$ dumbbells (which would be consistent with calling them small loops), they are better described as collections of $\langle 110 \rangle$ dumbbells, consistent with A15 phase. This view is supported by electronic structure calculations which find very similar energies for the different clusters, and Frenkel pair accumulation simulations which show the appearance of A15 clusters.

We acknowledge the Referee for the thorough reading of our manuscript and for her/his thought-provoking remarks and suggestions. We also appreciate that the Referee has accentuated our cutting-edge methodology based on distortion scores for defect detection. To the best of our knowledge, there are no other studies using methods of similar accuracy for the analysis of large-scale simulations. However, here we would like to emphasize that this accurate method is not a primary focus of the present paper. The distortion scores are employed as a tool to better describe the mechanism of dislocation loop formation via compact A15 clusters. This mechanism is different and more complex than the historically assumed scenario of direct accumulation of dumbbells into 2D loops.

Below we provide our point-by-point replies where we have addressed all the critical points and provided convincing arguments for the issues raised by the Referee. The response is divided into two parts: (2A) is focused solely on the scientific message, impact, and novelty of the present study; and (2B) addresses all the technical and methodological questions raised by the Referee.

2A. Scientific context, impact, novelty, and message of the present study

R2A-1 My biggest question to the authors, given that it has been submitted to Nature, is why this is an important distinction at such small sizes. The authors themselves acknowledge that it would be difficult to say whether black dots in TEM micrographs should be one cluster type or another, and that the energy barrier for dumbbell rotation or cluster transformation is order 0.5 eV, so is active at room temperature while inactive in their simulations. If the clusters were to grow large, as has been observed for C15 clusters in bcc iron, their sessile nature could well impact hardness. But at 7-8 interstitials is there any effect on material properties expected?

If there were a way to experimentally determine the nature of these clusters or their effect, this paper would clearly be worthy of Nature Comm.

We agree with the referee's assessment that the current discovery is outstanding and warrants experimental confirmation. We have followed the suggestion of the referee 1 that if the present discovery is true it should have a signature in the older resistivity recovery experiments. Furthermore, we have reinterpreted 50-year-old studies, which combine resistivity recovery and diffuse X-ray scattering experiments, in the context of recent *ab initio* calculations. Using accurate values of the *ab initio* elastic dipole tensors for various families of defects, we have reinterpreted these experimental studies, which were conducted in the late 1970s and early 1980s. Motivated by these experiments, we have included in the present study the third fcc metal, Ni. For all three investigated metals, we obtain a remarkable agreement with the experiments, and we can conclude that the unidentified 3D clusters evidenced by 50-year-old experiments are precisely the present A15 nano-phase clusters. Moreover, if in the case of Al and Cu the experiment indicates that 3D defects are lower than 7-8 SIAs, in the case of Ni the A15 clusters can have between 20 and 30 SIAs. Furthermore, in order to go beyond, the direct evidence in microscopy of A15 clusters is almost impossible due to their size. We have designed an experimental setup based on pure electron irradiated Ni that emphasizes a consequence of A15 clusters existence on large dislocation loops morphology: the A15 mechanism of loops formation yields a diverse population of Frank and perfect loops in pure Ni, not only the lowest energy one.

We summarize the main outcome of the present study: we have revised the 50-year-old formation mechanism of interstitial defects in fcc metals by pointing to the existence of a new genuine class of 3D defects in the form of A15 nano-phase aggregates. We have claim the existence of A15 nano-phase aggregates

and highlighted the major implications of this discovery. The implications, questions, and interrogations that arise are numerous and open valuable and rich perspectives for the community, which we are pleased to share.

We strongly believe that our prediction of compact Frank-Kasper nano-phase formation in fcc metals has fundamental importance for the Physics and Materials Science community, as it thoroughly revises the current scientific understanding of interstitial defect accumulation and dislocation loop formation in metals. We believe that the manuscript corrections, which include both additional theoretical developments and proper interpretation of existing experimental data, as well as experiments specially designed to indirectly acquire proof of the existence of A15 nano-phase clusters, will enable us to meet the criteria for high-quality research of the esteemed Nature Communications journal.

2B. Technical questions and answers

As it is, I have some technical questions below which might help the authors strengthen the theoretical argument made.

We thank the Referee for these questions and remarks, which have helped us to improve the manuscript.

R2B-1. Figures 3 and 4

R2B-1a. I wanted to find the relative importance of small A15 vs small loops by comparing the proportion of interstitials in each category, as line 190 states "The number of A15 clusters detected in this work per (MD) cascade is a small and represents only 4-5 % of interstitial atoms." while fig 4 suggests the number of A15 clusters decreases with dose (to 0.01dpa), and line 260 reads "No mixed structures of A15 with dislocation network were observed in our simulations"

We thank the Referee for this observation. The statements cited by the Referee describe our theoretical predictions for three different processes, not necessarily related. The line 190 points to an effect related to theoretical predictions in the present displacement cascades simulation, while Figure 4 (of the original submitted version) refers to Frenkel Pair Accumulation (FPA) simulations. Figures 3 and 4 (of the original submitted version) describe the results from FPA and no information from cascade simulations is provided on those Figures. Finally, the line 260 is a general conclusion regarding "mixed structures" not related to Figures 3 and 4.

Simulations of cascades and FPA are different methods that investigate various processes, therefore we have performed and analyzed both simulation types. While FPA allows for exploration of defects and processes dominated by short-range atomic rearrangements, cascades take into account long-range diffusion and recombination processes. Both simulations confirm the formation of A15 clusters and provide no evidence of mixed defect clusters where A15 coexist with dislocation lines within one defect cluster. These defects are referred to in the text as "mixed structures".

In order to avoid these confusions and improve the quality of the manuscript we have performed the following changes:

- To prevent misunderstanding between FPA and cascade simulations, we have moved the results of cascade simulations. Initially, in the originally submitted manuscript the cascade simulations were provided in Fig. 7 of the main text. We have decided to move them to **Supplementary Note 3** of the revised manuscript.
- Moreover, in the last version of the manuscript, we have reworded the sentence related to the "mixed structures" by adding the following phrase: **No defect configurations formed by A15 nano-phase clusters that are connected to dislocations or to dislocation networks were observed in our simulations.** before the section **On the role of A15 clusters in microstructural evolution.**

R2B-1b. If I am reading it correctly, Figure 3 shows the number of clusters in each category, not the number of interstitial atoms in each category. Figure 4 could give the number of interstitials in A15, but

not the number in the loops (we can't necessarily convert a line length into an area). Could the authors provide a simple comparison for the number of interstitials in each category, please?

The point addressed by the referee is important, and for this reason, we give it significant consideration in our analysis. The major point of this analysis is that reporting the number of interstitials is suitable for standard 2D SIAs clusters, but it is out of scope in the case of 3D clusters, as has been previously demonstrated in many studies concerning 3D clusters, such as [1, 22, 8, 23, 2, 3]. For this reason, we use the distortion score [24] that emphasizes the atoms in the A15 phase environment, which are far from the reference environment (pure fcc phase with Gaussian thermal noise, in the present case, see R2B-3b answer for more details).

For most geometries of 3D clusters, the exact number of SIAs cannot be extracted. For instance, a cluster of 19 A15 atoms can have from 10 to 13 SIAs. As Wigner-Seitz analysis for SIA detection is not reliable for 3D clusters like A15 and C15, we prefer not to convert the number of atoms in the A15 cluster in Fig. 4 into the number of SIAs in those clusters. Similarly, for the dislocation loops, conversion of line length into the exact number of SIAs will bring additional errors, as the precision can be at best $\pm 1 - 2$ SIA atoms even for small clusters with < 10 SIAs.

In order to avoid confusion and clarify this aspect, we have made the following changes:

- Because the number of atoms in A15 clusters is not the same as the number of SIAs, we have added in **Figure 1** the indication of the number of atoms together with the number of SIAs in A15 clusters.
- Furthermore, for clarity, we have changed the title of y axis in **Figure 4b** from “ A15 size” to “A15 size, number of atoms”.
- In addition to that, we have also added, when needed, in the section **Evidence of the A15 clusters formation from the large-scale calculations of radiation damage**, the number of SIAs that can contribute to the relevant defect size.

R2B-1c. In Figure 4 the histogram seems to start with A15 clusters of size 2. Is this a 100 dumbbell plus the octahedral interstitial? This seems very high energy- surely it would symmetrise to a triangular structure? Can the structure be shown in figure 2, please?

The histogram in Figure 4 (**Figure 5** in the revised version) starts with A15 configurations containing 3 atoms, which correspond to the most likely local atomic environments of A15. As a reference point, it can be a $\langle 100 \rangle$ dumbbell and octahedral interstitials, as well as one icosahedral face with 3 atoms. In FPA simulations, the occurrence of defect clusters does not necessarily follow the equilibrium statistics by occupying the most energetically favorable basin only; formation of higher energy A15 variants is possible.

R2B-1d. In Figure 4 the x-ranges seem to be different for A15 and loops. Is there a good reason, or can they be made the same?

The choice of range for x axis in **Figure 5** (Figure 4 in originally submitted paper) was governed by better visual representation of important information. In the case of A15 (**Figure 5b**), the x axis goes up to high doses in order to show how the size distribution of A15 clusters changes with dose. In the case of dislocation loops (**Figure 5c**), the important information (in the context of this study) is near the vertical dotted lines that indicated the limit of the regime where the microstructure is dominated by A15 clusters. As this limit is different for Al and Cu, the range of x axis **Figure 5c** is different for different metals.

R2B-1e. In Figure 4a) the small A15 clusters are shown. I can't really see how they relate to the clusters shown in Figure 1.

The clusters in **Figure 1** are the minimum DFT energy A15 configurations. In **Figure 5** (former Figure 4), the clusters are examples of commonly observed configurations from the FPA simulations (few among hundreds of different clusters). The structures from **Figure 1** and **Figure 5** belong to one family of defects, but they do not necessarily represent exactly the same configurations. For some small-size

clusters, it is possible to identify the correspondence (*e.g.* I_3^{A15} in **Figure 1** and the 5-atom cluster in **Figure 5a**), however, for relatively big clusters from FPA, it is not possible to unambiguously make the correspondence.

- In order to make the correspondence between the structures in **Figure 1** and **Figure 5a** more clear, we have added the indication of the number of atoms in A15 clusters in **Figure 1**.

R2B-1f. Line 126: "The distribution of A15 cluster sizes in Al have the shape of Poisson-like distribution". I didn't really understand why this should be so, and frankly can't see it myself in Figure 4b. (I tried to extract a couple of size-frequency histograms, they did not look particularly Poisson to me.) The authors should clarify why they think the probability of an injected interstitial joining a cluster is independent of the cluster size in the FPA method. If this can be done, it would be a useful contribution as it simplifies hugely the size-frequency histogram.

On a related note: line 152 states "the distribution of A15 cluster size [for Cu] does not follow a Poisson-like distribution" Was a chi-squared test done? Is this assessment carried out by eye?

The statement regarding the Poisson-like shape of the size distribution is related solely to the shape of the distribution and not to the fact that the counts are independent. Mathematically speaking, it is a general Gamma distribution. However, this point is far from the topic of this paper. For this reason, we have used the term "Poisson-like", which can be easily recognized by the broad audience of Nature Communications, rather than a cryptic formulation such as "particular one hump case of Gamma distribution".

The fact that distribution has a particular shape does not impact in any way the conclusion of this study. We have changed the two confusing propositions of the section **Evidence of the A15 clusters formation from the large-scale calculations of radiation damage**, into:

- The distribution of A15 cluster sizes in Al ... resembles the shape of a Poisson distribution with a maximum at low doses.
- The distribution of A15 cluster sizes in Cu is noisier than in Al.

R2B-2. 3D vs 2D interstitial clusters.

R2B-2a. In line 247 we read "A complete A15 icosahedron is built by 7 interstitial atoms", and in line 259 "Due to the important concentration of pressure within the cluster, A15 fully transform into loops". But a 7 interstitial loop also has 7 interstitials, so should have a comparable relaxation volume. If the authors want to make the claim that there is an elastic relaxation made possible by the transformation from A15 to loop, they should provide the dipole tensors or relaxation volumes.

We thank the referee for this remark. The Referee's affirmation that the relaxation volumes of 2D loops and 3D A15 clusters are comparable (see the table below) is indeed confirmed by our calculations. However, in the manuscript, we did not do any claims regarding the elasticity-driven mechanism of A15 cluster transformation. We addressed A15 \rightarrow loop transformation mechanism solely employing local physical quantities such as energy and pressure per atom (*e.g.* in Figure 6, Figure 9 and 3D cluster-related parts of the Discussion section (p.13-14) of the originally submitted manuscript). In Figure 9 (**Figure 8** of the revised manuscript) and the corresponding part of the Discussion section, we have only focused on the local pressure of i^{th} atom $p_i = \frac{1}{3V} \text{Tr}\sigma_i$. The important changes in the local pressure only manifest in the core regions of defects where elasticity (at least in linear form) can not be applied [11, 25, 14]. However, this is a nice perspective to study the aspects of non-linear elasticity, which is far beyond the goals of the present work.

Following the requests **R2B-2a** and **R2B-2b** of the Referee, we provide here in Table 1 the relaxation volume Ω^R of 1 and 7 SIAs in various configurations, as well as the relaxation volume tensor Ω_{ij} and the elastic dipole tensor P_{ij} of 7 SIAs extracted from the DFT simulations. The relaxation volume of 7 SIA clusters is, within some small fluctuations, nearly the same for the 3 types of investigated SIAs clusters. However, each of those relaxation volumes is smaller than the relaxation volume of 7 isolated SIAs. The

Table 1: Relaxation volumes Ω^R (in Ω_0 , zero pressure fcc atomic volume, units) for 1-SIA and 7-SIA clusters, relaxation volume tensors Ω_{ij} (in Ω_0 units) and elastic dipole tensors P_{ij} (in eV) for 7-SIA clusters. The values for 7-SIAs are extracted from DFT simulations performed in this work, the 1-SIA values are taken from Ref. [26]

	1-SIA Ω^R	7-SIA Ω^R	7-SIA Ω_{ij}	7-SIA P_{ij} (eV)
A15		10.80	$\begin{bmatrix} 3.60 & 0.00 & 0.00 \\ 0.00 & 3.60 & 0.00 \\ 0.00 & 0.00 & 3.60 \end{bmatrix}$	$\begin{bmatrix} 111.68 & 0.00 & 0.00 \\ 0.00 & 111.68 & 0.00 \\ 0.00 & 0.00 & 111.68 \end{bmatrix}$
Cu $\frac{1}{3}\langle 111 \rangle$	1.85 [26]	11.06	$\begin{bmatrix} 3.48 & 0.77 & 0.77 \\ 0.77 & 3.79 & 0.85 \\ 0.77 & 0.85 & 3.79 \end{bmatrix}$	$\begin{bmatrix} 108.01 & 23.90 & 23.90 \\ 23.90 & 117.47 & 26.39 \\ 23.90 & 26.39 & 117.47 \end{bmatrix}$
$\frac{1}{2}\langle 110 \rangle$	1.82 [26]	11.07	$\begin{bmatrix} 3.66 & 2.21 & 0.01 \\ 2.21 & 3.66 & 0.01 \\ 0.01 & 0.01 & 3.75 \end{bmatrix}$	$\begin{bmatrix} 113.56 & 68.43 & 0.24 \\ 68.43 & 113.56 & 0.24 \\ 0.24 & 0.24 & 116.21 \end{bmatrix}$
A15		11.76	$\begin{bmatrix} 3.92 & 0.00 & 0.00 \\ 0.00 & 3.92 & 0.00 \\ 0.00 & 0.00 & 3.92 \end{bmatrix}$	$\begin{bmatrix} 93.48 & 0.00 & 0.00 \\ 0.00 & 93.48 & 0.00 \\ 0.00 & 0.00 & 93.48 \end{bmatrix}$
Al $\frac{1}{3}\langle 111 \rangle$	2.51 [26]	12.42	$\begin{bmatrix} 4.14 & 0.93 & 0.93 \\ 0.93 & 4.14 & 0.93 \\ 0.93 & 0.93 & 4.14 \end{bmatrix}$	$\begin{bmatrix} 98.80 & 22.18 & 22.18 \\ 22.18 & 98.90 & 22.23 \\ 22.18 & 22.23 & 98.90 \end{bmatrix}$
$\frac{1}{2}\langle 110 \rangle$	2.46 [26]	12.39	$\begin{bmatrix} 4.32 & 1.72 & 0.07 \\ 1.72 & 4.32 & 0.07 \\ 0.07 & 0.07 & 3.75 \end{bmatrix}$	$\begin{bmatrix} 103.21 & 41.05 & 1.58 \\ 41.05 & 103.21 & 1.58 \\ 1.58 & 1.58 & 89.54 \end{bmatrix}$
A15		11.10	$\begin{bmatrix} 3.70 & 0.00 & 0.00 \\ 0.00 & 3.70 & 0.00 \\ 0.00 & 0.00 & 3.70 \end{bmatrix}$	$\begin{bmatrix} 146.74 & 0.00 & 0.00 \\ 0.00 & 146.74 & 0.00 \\ 0.00 & 0.00 & 146.74 \end{bmatrix}$
Ni $\frac{1}{3}\langle 111 \rangle$	1.86 [26]	11.18	$\begin{bmatrix} 3.44 & 0.82 & 0.82 \\ 0.82 & 3.87 & 1.08 \\ 0.82 & 1.08 & 3.87 \end{bmatrix}$	$\begin{bmatrix} 136.56 & 32.34 & 32.34 \\ 32.34 & 153.46 & 42.99 \\ 32.34 & 42.99 & 153.46 \end{bmatrix}$
$\frac{1}{2}\langle 110 \rangle$	1.87 [26]	11.35	$\begin{bmatrix} 3.77 & 2.61 & 0.08 \\ 2.61 & 3.77 & 0.08 \\ 0.08 & 0.08 & 3.81 \end{bmatrix}$	$\begin{bmatrix} 149.41 & 103.63 & 3.14 \\ 103.63 & 149.41 & 3.14 \\ 3.14 & 3.14 & 151.08 \end{bmatrix}$

difference between the relaxation volume of N SIAs and the relaxation volume of N SIA clusters can be important for multiscale studies, such as large-scale predictions of swelling.

In order to clarify the point raised by the referee, we have made the following changes:

- We have added the relaxation volume table in the **Supplementary Note 6**.
- In the **Discussion** of the manuscript, we have added a special section entitled **On the role of A15 clusters in microstructural evolution**, which highlights the particularities of the relaxation volume of traditional 2D dislocation loops and the 3D A15 nano-phase clusters, with the possible implications that should be addressed in future studies.

R2B-2b. On a related note: If there is a significantly different dipole tensor for the A15 and loop structures, then they should become preferentially formed in different strain states. How did the volume of the simulation cell change during the defect accumulation? Is it sufficient to tip the balance in favour of one or other structure? Could the uniaxial strain expected during ion irradiation or a TEM foil experiment lead to a different conclusion about the balance?

We thank the referee for raising these two important points: (i) the effect of the volume on the creation of various classes of defects as well as (ii) the effect of uniaxial strain in the selection of various morphologies of defects.

In order to investigate the effect of the volume of the simulation cell on the accumulation of defect structures, we have performed molecular MD simulations using NVT (fixed volume at $N \times \Omega_0$, where Ω_0 is the zero pressure atomic volume of fcc) and NPT (at zero pressure) simulations. In all these preliminary calculations we have noted that there is no significant variance in the results, indicating that there is no volume bias in the formation of A15 clusters and with respect to the other classes of clusters. Consequently, the volume changes do not have an impact on the conclusion of the present study.

Furthermore, the mechanism of formation of dislocation loops can bias the loop morphologies in materials under stress or in any strain field, such as thin foils or in the presence of a surface. The dipole tensor of A15 clusters has cubic symmetry (see the table in the reply to the question **R2B-2a**) and is isotropic without directional preferences, unlike that of dislocation loops [10, 11, 12, 13, 14]. This means that during the transformation A15 clusters \rightarrow dislocation loops, elastic interactions of the dislocation loop with the strain field will induce preferential planes of the habit plane of the planar loop defect, which corresponds to the lowest elastic interaction energy (for example the elastic energy accounted by the Eq. 28 of the review paper [14] or the treatment in the excellent general book for elasticity [15]). Consequently, the transformation mechanism of A15 clusters \rightarrow dislocation loops can influence the morphology of dislocation loops under strain, even at large sizes, which can be considerably biased. This issue raised by the referee provides great prospects for future studies and we appreciate the recognition of the importance of this study. We have made the following changes:

- In the section **Methods: Large scale calculations of radiation damage**, in the end, we have added the following sentence: We have performed NVT and NPT MD simulations and we found no significant variance in the results, indicating that there is no volume bias in the formation of A15 clusters and with respect to the other classes of clusters.
- In the section **Discussion: On the role of A15 clusters in microstructural evolution** we have added the following paragraph: Finally, the present mechanism of dislocation loops formation via A15 clusters can bias the loop morphologies in the microstructure of materials under stress or in any strain field, such as thin foils or in the presence of a surface. The dipole tensor of A15 clusters has cubic symmetry (see Supplementary Note 6), *i.e.* it has no directional preferences, unlike that of dislocation loops [10, 11, 12, 13, 14]. This means that during the transformation A15 clusters \rightarrow dislocation loops, elastic interactions of the dislocation loop with the strain field will induce preferential planes of the habit plane of the planar loop defect, which correspond to the lowest elastic interaction energy [14, 15]. Consequently, the mechanism of transformation A15 clusters \rightarrow dislocation loops can impact the morphology of dislocation loops under strain, even at large sizes, which can be considerably biased.

R2B-2c. Line 259: "A15 fully transform into loops. No mixed structures of A15 with dislocation network were observed in our simulations". Is this consistent with figures 3c),3d), and 4b)? There seem to be plenty of A15 structures in these figures.

Indeed, Figures 3c, 3b, and 3d (of the initially submitted version of the manuscript) contain dislocation loops and A15 clusters. However, for each defect, there is either a "pure" loop/dislocation or A15 cluster. There are no structures where A15 clusters and dislocation loops are connected and coexist within the same cluster defect. Such defects called "mixed" structures were observed for C15 clusters in bcc Fe (see for instance Figure 6 and Figure 8 in Ref. [8]) and the discussion in end of the section **3D interstitial clusters in fcc and bcc lattices: A15 versus C15**. However, in contrast to C15, we have observed in

the present MD simulations in Al and Cu that A15 clusters fully transform into dislocation loops without forming “mixed” configurations. This peculiar behaviour of A15 *versus* C15 clusters is described at the end of section **3D interstitial clusters in fcc and bcc lattices: A15 versus C15**.

In order to avoid confusion we have replaced the term “mixed configurations” at the end of the section **3D interstitial clusters in fcc and bcc lattices: A15 versus C15**. We have reformulated the same phrase as it is mentioned also the answer **R2B-1a**. Moreover, in order to focus on the main message of the present **Figure 4**, *i.e.* the formation of 3D nano-phase prior to dislocation loops, we have moved the graphical visualization of SIAs analysis of FPA simulations to **Supplementary Note 3**.

R2-2d. Line 284: “These observations [of 10dpa Al at 400K] are in agreement with our DFT and FPA simulations”. Consistent perhaps, but not really agreement. The FPA simulations are 0.01 dpa at very low temperature.

We thank the Referee for this remark. The sentence has been rephrased according to the Referee’s suggestion.

R2-2e. Line 418: I am concerned by the statement “atoms are randomly displaced in such a way that their distance from any other atoms is not smaller than 1.4 Å.”. I understand that this has computational benefits, in that a possible singularity in the potential energy surface is avoided, and large forces are not generated which could perhaps start a displacement cascade. But surely this cutoff also introduces biases. Firstly, the higher the cutoff length, the more it will tend to make interstitial insertion preferentially into open volumes, and so into lower energy positions. This could have the effect of promoting the growth of one type of interstitial cluster over another. As this is the main theme of the paper, this possibility should not be lightly ignored. Secondly, as the cutoff is apparently the same for both Cu and Al, elements with rather different lattice parameters, the effect of selecting one interstitial position over another for Cu and Al changes, perhaps giving a different or exaggerated conclusion. As a (very rough) estimate, I find that the ratio of acceptable volumes for tetra:octa positions in pristine fcc lattice is 2.5:1 for Cu and 1:1 for Al. I can’t say if this will distort the findings, but a convergence study should be performed to confirm it does not have an effect.

The referee has raised an interesting question on the very details of FPA simulations. He/she is right that setting a cutoff distance for interstitial insertions has a computational benefit, not only to avoid any singularity that could arise from very short distances between the inserted atoms and the lattice but also to prevent any “small” displacement cascade to occur. Another benefit is that it minimizes the inserted potential energy that transforms into kinetic energy (and therefore the temperature) during the relaxation. This helps the thermostat to handle the heating of the box and reduces the time required to bring the system back to the target temperature.

However, we did not evidence detectable bias that could be introduced by the choice of the cutoff. We have done FPA with different cutoffs, namely 1.0, 1.2, and 1.4 Å in both Al and Cu. The results obtained do not differ from each other. Aluminium sees the nucleation of mainly Frank loops while copper sees a majority of prismatic loops. This might be surprising since the interstitial rooms in aluminium and copper are quite different and could produce different biases as observed by the referee. However, appropriate explanations can be provided to clarify this apparent contradiction.

It is true that open volumes could appear favoured when operating the insertion of interstitials since the bigger they are, the higher the probability to be filled. It is true that some interstitial positions could seem favoured with respect to others. Nevertheless, both rationales stand only when considering static and uncorrelated situations. They do ignore the oversaturation of vacancies and interstitials produced by FPA and they do ignore the dynamic processes that produce gathering of interstitials on one hand and of vacancies on the other, and also their mutual recombination (annealing of Frenkel pairs).

In fact, at each stage of Frenkel pairs creation, atoms chosen randomly are displaced to sites that are also randomly chosen. The random choice of atoms to be displaced from their sites does concern all atoms, including atoms at the edge of formed voids, atoms in interstitial positions, atoms in clusters and

even atoms being part of loops. Hence, voids can grow from this process; interstitial clusters as well as interstitial loops can be destroyed. Similarly, the random introduction of atoms could as well fill voids or contribute to the creation of clusters of any type. Obviously, the diversity of static situations makes it difficult to evaluate the balance between all processes, but no bias seems to drive the system toward one side or the other.

Conversely, the dynamic of each interstitial and vacancy created has to be considered to discern the driving process of the creation of voids, clusters and loops of any type, during the MD simulations. Indeed, each interstitial (vacancy) interacts with defects around, and depending on the situation will dynamically feed or destroy point/extended defects like loops, vacancies, clusters or single interstitials. The emblematic example of such a process is the recombination of Frenkel pairs: single interstitial and single vacancy recombine when their distance is lower than the recombination radius according to standard rate equations. Here, the recombination is not instantaneous: it does occur within a few picoseconds at a moderate distance. Similar dynamics interactions do exist for all interactions of defects, most of them being possible at short range, *i.e.* around 10 Å.

The following sentence has been added to the section **Methods: Large scale calculations of radiation damage**:

- We tested also different cutoffs, namely 1.0, 1.2, and 1.4 Å, and the results obtained do not differ from each other.

R2B-3. Detection of A15 clusters in fcc.

R2B-3a. As the classification of clusters is vital to this work, it is important that it is robust. My concern is that a very small cluster of 3-4 interstitials, which people have previously not really known how to correctly label, is being called A15 structure with high confidence here. From what I have seen, I believe the authors' new "distortion score" method is a good way of discriminating structures. The question mark I have comes from line 445: "its proximity to the learned A15 structures", which might suggest that the algorithm has been told to classify small clusters as A15. Was a proximity to "learned loop structures" considered? I note in line 133 "many interstitial Frank loops appear at the size of 7-8 interstitial atoms". Is it possible for a 3-4 interstitial cluster to be labelled a Frank loop? Were any 3-4 interstitial clusters labelled anything other than A15?

The referee has right: high throughput screening of structural defects in massive molecular dynamics simulations is of primary importance topic in modern physics, especially in situations where the production of defects gives rise to unexpected and diverse morphology. We also thank the referee for finding our methodology interesting and a good way to handle this problem.

The present paper uses the distortion score [24] that enables a high level of accuracy in the characterization of the morphology of defects in crystalline materials. This technique is based on the statistical analysis of the molecular dynamics trajectories in the descriptor space and isolation, discrimination, and identification of defects as outlier instances of that distribution. The main point, and likely source of confusion, is that the distortion score technique classifies atoms by introducing a statistical distance *per atom* and not per cluster. These statistical distances *per atom* are then used to cluster atoms with similar distances and which are in geometrical proximity. So the main quantity is a positive real number that embeds the atomic environment of the i^{th} atom $\mathbf{q}_i \in \mathbb{R}^{3n_i}$, where n_i is the number of neighbouring atoms of that i atom within a cut-off distance. Consequently, when we speak about 3 or 4 atoms with an A15 structure, it means that the statistical distances associated with those atoms are close to the statistical distances of the atoms that form a perfect A15 cluster. This is the main outcome of the present way of reinterpreting the nature of defects, as was extensively presented in reference [24]. Using this technique, there is a very high probability that the clusters built by the atoms identified as A15 cannot belong to another family of defects.

R2B-3b. I see "noise with normal distribution $N(0;0.08A)$ was applied on the perfect atomic positions", which seems fine for Cu at 300K. But Al has a larger Debye-Waller factor. Should the same value be used? Does it matter? What account is made for local strain? Do the possibly large local strains have

an impact for these dense microstructures?

The distortion score describes a statistical distance from the center of a reference distribution in the feature space of atomic descriptors. In the present study, the reference distribution was constructed by applying noise with a normal distribution $\mathcal{N}(\mu = 0; \sigma = 0.08 \text{ \AA})$ to relaxed configurations of A15 clusters embedded in fcc matrix of Al and Cu. Most commonly, a σ of 0.06-0.08 \AA is a good choice for the analysis of low-temperature simulations (our FPA is performed at 300 K) of metallic systems, as seen in Cu, Al, Ni, Fe, and W systems. Probably, an optimal value is the fast estimation of root mean square displacements of atoms at temperature T as $\frac{1}{2\pi T_D} \sqrt{\frac{2T}{m}}$, where T_D and m are the Debye temperature and mass, respectively. In the case of Al, Cu, and Ni it yields 0.08 \AA , 0.06 \AA and 0.05 \AA respectively. For higher temperatures, we advise scaling the displacement amplitude of the reference structures with that of the temperature of interest and performing benchmark tests on defect detection. Here, this is not the case; the present analysis is not biased by the amplitude of normal noise applied to perfect structures, as long as the amplitude of the noise is within the indicated above limits. The main reason for this particular case is that the distortion score distances of the A15 phase are very far from the distortion score associated with the fcc bulk distribution.

In order to correctly account for the strain field of A15 clusters, we have generated reference structures from A15 clusters embedded in fcc with a_0^{fcc} . Thus, the detection procedure recognizes that A15 is in a constrained environment relevant to these interstitial clusters.

R2B-3c. On a related note for Figure 4: How was it determined if an interstitial cluster $N < 10$ was Frank or Prismatic? How was the loop perimeter determined for small clusters < 10 interstitials? Ovito's DXA method generally fails at small size because there is no Burgers circuit through perfect material. The authors should consider how to make the argument compelling that the symmetry of the small clusters is that of $\langle 100 \rangle$ dumbbells not $\langle 110 \rangle$ or $\langle 111 \rangle$, and how they define loops $N < 10$.

We thank the referee for this comment. The detection of small interstitial clusters is indeed challenging. The $\frac{1}{3}\langle 111 \rangle$ and $\frac{1}{2}\langle 110 \rangle$ clusters with $N \geq 5$ SIA are still possible to detect by the DXA and the circumference of those small loops with 5-6 SIA appears in former **Figure 4c** as the smallest size loops.

R2-3d. As a general note, there are too many grammatical errors and spelling mistakes in the manuscript. Poorly proof-read lines like 128 or 432 should not make it to referees: "This distribution is not biased by attractive configuration and sample a pure metastable basin" "... with a transition to the equilibrium potential between 1.7 to the equilibrium potential between 1.7 and 2.0A and 2.0A."

We thank the referee for this remark. In the last submitted version of the paper, we have performed careful proofreading of the manuscript.

Reviewer #3

This manuscript presents interesting computational results that suggest the initial stages of point defect clustering in FCC metals may be more complex than historically assumed. Overall, the paper is well written and the authors use appropriate caution regarding the general validity of the results. I have a few minor comments.

We thank the Referee for the careful reading of our manuscript and recognition of the impact of this study on the historical vision of defect formation processes in fcc metals. We acknowledge the Referee for the raised points that will certainly help to enhance the message of the paper and improve its clarity. Below we provide our point-by-point replies.

R3-1. The authors should consider some alternative wording in lines 37-40 (p. 2). In line 37, the statement “defects... aggregate in voids that form SFTs” implies that 3D voids are first formed and then transform into SFTs, which is contrary to the mainstream understanding of vacancy cluster evolution; in general, it is extremely rare for void-SFT transformation (or vice versa) to occur, due to the enormous local atomic rearrangement that is required for vacancy cluster sizes exceeding 10 defects. Line 38 implies that the vacancies invariably first cluster into a 3D void before collapsing into a planar loop; this is contrary to the current conventional thinking where loops are formed via one-by-one addition of vacancies, as noted elsewhere in the manuscript. In addition to the modeling studies that have predicted SFT formation from vacancy loops (lines 38-39), there are numerous in-situ TEM studies (*e.g.*, Matsukawa, Science vol. 318 (2007) p.959, etc.) that have observed vacancy loop conversion to SFT. Reports claiming conversion of SFTs to voids (refs. [15, 19] discussed on p. 2, line 37 of the manuscript need further confirmation (there are far more numerous experimental and computational studies that have not observed such a conversion).

We thank the Referee for pointing out an unclear description of the process of vacancy cluster growth in fcc crystals that was given in the Introduction. We fully agree that the initial formulation was ambiguous and caused confusion. As the Referee points out, the misunderstanding comes from line 37 of the submitted version. Indeed, the SFTs are formed mostly after the accumulation of vacancies, after one-by-one accumulation in planar/sessile structures that subsequently transform into SFT. In order to improve the clarity, we have reworded the paragraph in question.

The corrections appear at the end of **page 2**, of the revised manuscript:

In the case of vacancies, the elementary defects diffuse and aggregate in voids and stacking fault tetrahedra (SFTs) [27, 28, 29]. Many experimental and theoretical studies [30, 28, 31, 32], indicate that planar vacancy clusters rearrange into SFTs by assembling $\frac{1}{3}\langle 111 \rangle$ Frank partial dislocations. Other works, based on numerical simulations, suggest that voids can directly transform into SFTs [33, 34].

R3-2. The introduction section discusses both vacancy and interstitial defect clusters, whereas the modeling results only examine interstitial cluster evolution. Is the A15 cluster mechanism considered to potentially also occur in vacancy cluster evolution?

We thank the Referee for this question. Indeed, our work is focused on purely interstitial clusters and their nucleation through the A15 Frank-Kasper nano-phase. Understanding the influence of vacancies on the evolution of A15 clusters in the microstructure is not straightforward, therefore this question deserves a thorough investigation. However, this subject is far beyond the scope of the present study and deserves to be considered as a separate subject in future studies. In our opinion, there are at least three major directions that should be addressed by the community:

- How the kinetic pathways of vacancies are driven by elastic field and the interaction of A15 with vacancies?
- Does the long-range bias induced by the A15 strain field act differently on the diffusion of migrating interstitials and vacancies? This asymmetry in diffusion can impact the growth mechanism of

clusters that potentially can impact the microstructural evolution.

- The A15 nano-phases considered in this work are very dense and compact structures built exclusively by atoms in interstitial positions in an fcc matrix. In perspective, it is worth investigating if there are any mixed structures that combine vacancies and interstitials, as in the case of C15 clusters embedded into bcc matrix [8].

Those questions emphasize the important implications of the concepts presented in this paper and open up many perspectives for the community. In order to highlight these perspectives, we have added an entire section **Discussion: On the role of A15 clusters in microstructural evolution**.

R3-3. The author definition for the term “loop size” should be indicated in the text; does this mean radius, diameter, or loop circumference? On p. 8, line 132 the manuscript states that a loop size of 2.0 to 2.1 nm corresponds to 7-8 atoms, which seems nonphysical if “size” means radius or diameter. Please recheck.

We thank the Referee for this comment. The “loop size” in this work refers to the length of the dislocation line that encircles 2D defect clusters. We agree that this notion of “size” is very specific for the atomic-scale simulations and can be confusing for the broad audience of Nature Communications. To improve the clarity, we have reformulated the relevant parts of the text of the section **Results: Evidence of the A15 clusters formation from the large-scale calculations of radiation damage** and have corrected the name of y axis in **Figure 5c** (former Figure 4c) from “loop size” to “loop circumference”.

R3-4. On p. 19 (Methods), the potential for artifact(s) associated with the extremely high damage rate (3×10^5 dpa/s) for the FPA simulations that will introduce extremely high defect supersaturations should be briefly noted.

The referee brings up an important point that was not well emphasized in the manuscript. In this work, we employ Frenkel Pair Accumulation (FPA) as a tool to explore the complex energetic landscape of defects under irradiation. We are particularly interested in the onset of SIA cluster formation and investigations of the small cluster “zoology”. Therefore, we only focus on the early stages of irradiation and we use large simulation cells where defects are still not saturated at the moment of formation of small dislocation loops. As mentioned by the Referee, at later stages of simulations, defects will supersaturate and possibly create some artifacts, such as an extremely dense dislocation network, that do not necessarily correspond to any realistic microstructure. Moreover, due to the high irradiation flux in FPA simulation, establishing the direct link with the kinetic effects based on these simulations is impractical and we do not intend to do this in this work.

In the originally submitted manuscript, the utility of FPA simulations was only briefly mentioned in the Methods section. In order to better emphasize the role of the method in our study, we have added explanations to the relevant part of the section **Results: Evidence of the A15 clusters formation from the large-scale calculations of radiation damage**:

Simulations of Frenkel pair accumulation (FPA) is particularly suitable for extensive exploration of defect populations, as well as of the related processes dominated by short-range diffusion [35, 8, 36]. Here we employ FPA as a tool to explore the onset of SIA cluster formation and the appearance of first dislocation loops. To complement the FPA simulations, we also perform calculations of displacement cascades, as they take into account long-range, *i.e.* thermal, diffusion processes (see Methods). The simulations in fcc Al and Cu are performed using EAM potentials that are numerically fast and allow for good qualitative agreement with DFT calculations (see Supplementary Note 1). For fcc Ni, there is no suitable semi-empirical potential, therefore, no large-scale simulations were performed for Ni.

Additional description of the FPA method has been added to the section **Methods: Large scale calculations of radiation damage** :

In order to mimic electron irradiation, the periodic creation of Frenkel pairs was first introduced by Limoge et al. [35] and has been further adapted for studies of irradiation-induced defects in different materials either at finite temperature or 0 K [8, 36]. Due to the high irradiation flux (more than 10^6

dpa/s) typical for FPA, establishing the direct link with the kinetic effects based on these simulations is impractical. Nonetheless, the method is particularly useful for exploring the morphology of defects and their density under irradiation. For example, the results of a high-flux FPA simulation can be applied to lower-flux overlapping cascades [36].

R3-5. In the methods section (discussion of Eq. 1) or elsewhere (introduction?), in addition to the Hirth and Lothe reference the authors should consider citing one of the numerous journal articles that have calculated the relative energies of perfect and faulted loops, *e.g.*, Kroupa Czech J. Phys B 1960, Zinkle et al. Phil Mag A 1987, or Povstenko J. Phys. D 1995.

We thank the referee for this remark. In the latest version of the manuscript, we cite the suggested studies in the **Introduction**.

R3-6. Several minor typos should be corrected in the manuscript.

We thank the Referee for indicating these typos.

- p. 2, line 49: “other” → “others” Done
- p. 3, line 71: “octahedral” → “octahedra” Done
- Fig. 3 caption (3rd line): “Microstrucure” → “Microstructure” Done
- p. 7, line 125: duplicate word “over over” → “over” Done
- p. 12, line 228: “dislocations” → “dislocation” Done
- p. 13, line 249: “then” → “than” Done

References

- [1] M.-C. Marinica, F. Willaime, and J.-P. Crocombette. Irradiation-induced formation of nanocrystallites with C15 laves phase structure in bcc iron. *Phys. Rev. Lett.*, 108:025501, 2012.
- [2] R. Alexander, M.-C. Marinica, L. Proville, F. Willaime, K. Arakawa, M. R. Gilbert, and S. L. Dudarev. Ab initio scaling laws for the formation energy of nanosized interstitial defect clusters in iron, tungsten, and vanadium. *Phys. Rev. B*, 94(2):024103, 2016.
- [3] Rebecca Alexander, Laurent Proville, Charlotte S. Becquart, Alexandra M. Goryeava, Julien Deres, Clovis Lapointe, and Mihai-Cosmin Marinica. Interatomic potentials for irradiation-induced defects in iron. *J. Nucl. Mater.*, 535:152141, 2020.
- [4] P. Ehrhart, P. Jung, H. Schultz, and H. Ullmaier. *Atomic Defects in Metals*. Springer-Verlag, Berlin, 1991.
- [5] L. Liu, N. Gao, Y. Chen, R. Qiu, W. Hu, F. Gao, and H. Deng. Formation mechanism of $\langle 111 \rangle$ interstitial dislocation loops from irradiation-induced C15 clusters in tungsten. *Phys. Rev. Mater.*, 5:093605, 2021.
- [6] A. Esfandiarpour, J. Byggmästar, J. P. Balbuena, M. J. Caturla, K. Nordlund, and F. Granberg. Effect of cascade overlap and C15 clusters on the damage evolution in Fe: An OKMC study. *Materialia*, 21:101344, 2022.
- [7] Jie Gao, Ermile Gaganidze, and Jarir Aktaa. Relative population of $1/2\langle 111 \rangle$ and $\langle 100 \rangle$ interstitial loops in alpha-Fe under irradiation: Effects of C15 cluster stability and loop one-dimensional movement. *Acta Materialia*, 233:117983, 2022.
- [8] A. Chartier and M. C. Marinica. Rearrangement of interstitial defects in alpha-Fe under extreme condition. *Acta Mater.*, 180:141–148, 2019.
- [9] J. P. Balbuena, M. J. Aliaga, I. Dopico, M. Hernández-Mayoral, L. Malerba, I. Martin-Bragado, and M. J. Caturla. Insights from atomistic models on loop nucleation and growth in α -Fe thin films under Fe+ 100keV irradiation. *J. Nucl. Mater.*, 521:71–80, 2019.
- [10] S. L. Dudarev, R. Bullough, and P. M. Derlet. Effect of the $\alpha - \gamma$ phase transition on the stability of dislocation loops in bcc iron. *Phys. Rev. Lett.*, 100:135503, 2008.
- [11] E. Clouet. Elastic energy of a straight dislocation and contribution from core tractions. *Philos. Mag.*, 89:1565, 2009.
- [12] C. Varvenne, F. Bruneval, M.-C. Marinica, and E. Clouet. Point defect modeling in materials: Coupling ab initio and elasticity approaches. *Phys. Rev. B*, 88(13), 2013.
- [13] C. Varvenne, O. Mackain, and E. Clouet. Vacancy clustering in zirconium: An atomic-scale study. *Acta. Mater.*, 78:65377, 2014.
- [14] Emmanuel Clouet, Céline Varvenne, and Thomas Jourdan. Elastic modeling of point-defects and their interaction. *Computational Materials Science*, 147:49–63, 2018.
- [15] Toshio Mura. *Micromechanics of Defects in Solids (Mechanics of Elastic and Inelastic Solids, 3)*. Springer; 2nd edition (November 30, 1987), New York, 1987.
- [16] P. Ehrhart and W. Schilling. Investigation of interstitials in electron-irradiated aluminum by diffuse-X-ray scattering experiments. *Physical Review B*, 8(6):2604–2621, 1973.
- [17] J. B. Roberto, B. Schoenfeld, and P. Ehrhart. Investigation of interstitial clustering in Al following electron irradiation at low temperature. *Physical Review B*, 18:2591–2597, 1978.
- [18] P. Ehrhart and R. S. Averback. Diffuse X-ray scattering studies of neutron- and electron-irradiated Ni, Cu and dilute alloys. *Philosophical Magazine A*, 60:283–306, 1989.

- [19] O Bender and P Ehrhart. Self-interstitial atoms, vacancies and their agglomerates in electron-irradiated nickel investigated by diffuse scattering of x-rays. *Journal of Physics F: Metal Physics*, 13(5):911, 1983.
- [20] R. S. Averback and P. Ehrhart. Diffuse X-ray scattering studies of defect reactions in electron-irradiated dilute nickel alloys. I. Ni-Si. *Journal of Physics F: Metal Physics*, 14:1347, 1984.
- [21] O. Bender and P. Ehrhart. Self-interstitial atoms, vacancies and their agglomerates in electron-irradiated nickel investigated by diffuse scattering of X-rays. *Journal of Physics F: Metal Physics*, 13(5):911, 1983.
- [22] Julien Dérès, Laurent Proville, and Mihai-Cosmin Marinica. Dislocation depinning from nano-sized irradiation defects in a bcc iron model. *Acta Mater.*, 99:99, 2015.
- [23] L. Dézerald, M.-C. Marinica, L. Ventelon, D. Rodney, and F. Willaime. Stability of self-interstitial clusters with C15 Laves phase structure in iron. *J. Nucl. Mater.*, 449:219, 2014.
- [24] A. M. Goryaeva, C. Lapointe, C. Dai, J. Dérès, J.-B. Maillet, and M.-C. Marinica. Reinforcing materials modelling by encoding the structures of defects in crystalline solids into distortion scores. *Nat. Commun.*, 11:4691, 2020.
- [25] E. Clouet, L. Ventelon, and F. Willaime. Dislocation core energies and core fields from first principles. *Phys. Rev. Lett.*, 102:055502, 2009.
- [26] P.-W. Ma and S. L. Dudarev. Nonuniversal structure of point defects in face-centered cubic metals. *Phys. Rev. Materials*, 5:013601, 2021.
- [27] M. Kiritani. Story of stacking fault tetrahedra. *Mater. Chem. Phys.*, 50(2):133 – 138, 1997.
- [28] Y. Matsukawa and S. J. Zinkle. One-Dimensional Fast Migration of Vacancy Clusters in Metals. *Science*, 318(5852):959–962, 2007.
- [29] R. Schibli and R. Schäublin. On the formation of stacking fault tetrahedra in irradiated austenitic stainless steels – a literature review. *J. Nucl. Mater.*, 442:S761–S767, 2013.
- [30] J. Silcox and P. B. Hirsch. Direct observations of defects in quenched gold. *Philos. Mag. A*, 4(37):72–89, 1959.
- [31] B. D. Wirth, V. Bulatov, and T. Diaz de la Rubia. Atomistic simulation of stacking fault tetrahedra formation in Cu. *J. Nucl. Mater.*, 283-287:773–777, 2000.
- [32] J. P. Hirth and J. Lothe. *Theory of dislocations*. Wiley, New York, 1982.
- [33] B. P. Uberuaga, R. G. Hoagland, A. F. Voter, and S. M. Valone. Direct Transformation of Vacancy Voids to Stacking Fault Tetrahedra. *Phys. Rev. Lett.*, 99(13):135501, 2007.
- [34] H. Wang, D. Rodney, D. Xu, R. Yang, and P. Veyssière. Pentavacancy as the key nucleus for vacancy clustering in aluminum. *Phys. Rev. B*, 84(22):220103, 2011.
- [35] Y. Limoge, A. Rahman, H. Hsieh, and S. Yip. Computer simulation studies of radiation induced amorphization. *J. Non-Cryst. Solids*, 99(1):75–88, 1988.
- [36] P. M. Derlet and S. L. Dudarev. Microscopic structure of a heavily irradiated material. *Phys. Rev. Materials*, 4:023605, 2020.

REVIEWERS' COMMENTS

Reviewer #2 (Remarks to the Author):

Where the previous draft felt like a computational observation, this one is a fully rounded scientific proposition of significance. The authors have fully answered my questions, and more. I have no hesitation in recommending this paper for publication in Nature Comms.

Reviewer #3 (Remarks to the Author):

The authors have made numerous modifications to the draft manuscript in response to the referee comments. These changes have substantially improved the manuscript value. In particular, the detailed assessment of experimental isochronal annealing results (electrical resistivity and diffuse x-ray scattering) was in important addition to the manuscript. I do not have additional suggested changes.

Response reviewers

Ref. No.: NCOMMS-21-39664A-Z

Title: Compact A15 Frank-Kasper nano-phases at the origin of dislocation loops
in face-centred cubic metals

Authors: Alexandra M. Goryaeva et al.

April 24, 2023

Reviewer #2

Where the previous draft felt like a computational observation, this one is a fully rounded scientific proposition of significance. The authors have fully answered my questions, and more. I have no hesitation in recommending this paper for publication in Nature Comms.

We thank the Referee for thorough review of our manuscript and for the recognition of our work. Moreover, all her / his suggestions through the review process have significantly improved the manuscript and we express our sincere gratitude for this constructive criticism.

Reviewer #3

The authors have made numerous modifications to the draft manuscript in response to the referee comments. These changes have substantially improved the manuscript value. In particular, the detailed assessment of experimental isochronal annealing results (electrical resistivity and diffuse X-ray scattering) was in important addition to the manuscript. I do not have additional suggested changes.

We thank the Referees for the careful reading of our manuscript. We fully agree that providing solid argumentation based on the experimental observations enabled significant improvement of the paper, making it appropriate for the wide readership of Nature Communications.